

# Enhancing the classification of isolated theropod teeth using machine learning: a comparative study

Carolina S. Marques[1], Emmanuel Dufourq[2,3,4], Soraia Pereira[1,5], Vanda F. Santos[6,7,8] and Elisabete Malafaia[8,9]

[1] Centro de Estatística e Aplicações, Faculdade de Ciências, Universidade de Lisboa, Lisboa, Portugal
[2] African Institute for Mathematical Sciences, Muizenberg, South Africa
[3] Department of Mathematical Sciences, University of Stellenbosch, Stellenbosch, South Africa
[4] African Institute for Mathematical Sciences Research and Innovation Centre, Kigali, Rwanda
[5] Departamento de Estatística e Investigação Operacional, Faculdade de Ciências, Universidade de Lisboa, Lisboa, Portugal
[6] Departamento de Geologia, Faculdade de Ciências, Universidade de Lisboa, Lisboa, Portugal
[7] PaleoIbérica Research Group, University of Alcalá, Alcalá de Henares, Spain
[8] Instituto Dom Luiz, Faculdade de Ciências, Universidade de Lisboa, Lisboa, Portugal
[9] Grupo de Biología Evolutiva, Universidad Nacional de Educación a Distancia, Madrid, Spain

Corresponding author
Carolina S. Marques,
csmarques@fc.ul.pt

## ABSTRACT

Classifying objects, such as taxonomic identification of fossils based on morphometric variables, is a time-consuming process. This task is further complicated by intra-class variability, which makes it ideal for automation via machine learning (ML) techniques. In this study, we compared six different ML techniques based on datasets with morphometric features used to classify isolated theropod teeth at both genus and higher taxonomic levels. Our model also intends to differentiate teeth from different positions on the tooth row (*e.g.*, lateral, mesial). These datasets present different challenges like over-representation of certain classes and missing measurements. Given the class imbalance, we evaluate the effect of different standardization and oversampling techniques on the classification process for different classification models. The obtained results show that some classification models are more sensitive to class imbalance than others. This study presents a novel comparative analysis of multi-class classification methods for theropod teeth, evaluating their performance across varying taxonomic levels and dataset balancing techniques. The aim of this study is to evaluate which ML methods are more suitable for the classification of isolated theropod teeth, providing recommendations on how to deal with imbalanced datasets using different standardization, oversampling, and classification tools. The trained models and applied standardizations are made publicly available, providing a resource for future studies to classify isolated theropod teeth. This open-access methodology will enable more reliable cross-study comparisons of fossil records.

## INTRODUCTION

Isolated theropod teeth are commonly found in fossil sites worldwide (*e.g.*, *Hendrickx, Mateus & Araújo, 2015*; *Malafaia et al., 2017*; *Hendrickx et al., 2019*) due to their greater resistance to chemical alterations and abrasion, compared to other osteological remains (*Argast et al., 1987*), as well as the continuous replacement of the dentition throughout the animal lives (*Hanai & Tsuihiji, 2018*; *D'Emic et al., 2019*). In contexts where other skeletal remains are sparse or entirely absent, isolated theropod teeth can provide valuable insights for the knowledge of faunal diversity and ancient ecosystems (*Hendrickx et al., 2019*). The emergence of computational programs has significantly enhanced the study of these fossil remains through both qualitative and quantitative methods.

Despite their abundance, classifying isolated theropod teeth is still challenging due to the poorly documented distribution of dental features for most taxa and the high morphological convergence in the dentition of distantly related theropods with similar feeding strategies (*e.g.*, *Smith, Vann & Dodson, 2005*; *Hendrickx et al., 2019*). Morphometric tools for the study of theropod teeth were firstly approached by *Farlow et al. (1991)* and later developed by *Smith, Vann & Dodson (2005)* and *Smith (2007)*. These authors compiled a large dataset of morphometric variables based on crown and denticle measurements for different theropod taxa, which have been subsequently extended on later works, both in number of specimens and dental-based variables (*e.g.*, *Smith, 2007*; *Larson & Currie, 2013*; *Hendrickx, Mateus & Araújo, 2015*; *Gerke & Wings, 2016*; *Hendrickx, Tschopp & Ezcurra, 2020*; *Isasmendi et al., 2024*). Therefore, traditional morphometric approaches, such as principal component analysis (PCA), discriminant function analysis (DFA), and canonical variate analysis (CVA), combined with cladistic methods based on dentition characteristics have been widely used in the last decades to support the taxonomic identification of isolated theropod teeth. These methods are valuable approaches to distinguish between different theropod taxa (*Hendrickx et al., 2019*). However, conventional morphometric methods often encounter difficulties when dealing with imbalanced data, which is an inherent issue in paleontological datasets (*e.g.*, *Wills, Underwood & Barrett, 2020*; *Ballell, Benton & Rayfield, 2022*). Also, data imbalance, due to over-representation of certain taxa and under-representation of others can bias classification results, undermining the accuracy and reliability of these methods (*Alonso et al., 2018*; *Averianov, Ivantsov & Skutschas, 2019*). Class imbalance is a significant challenge in machine learning since it results in model overfitting and can negatively affect the overall performance of the model. When faced with imbalanced data, models often become biased toward the majority class, which leads to overtraining on these samples. This skewness in the class distribution can result in high overall accuracy that is often misleading, as the model will most likely misclassify the minority class, which tends to be overlooked (*Mujahid et al., 2024*). Undersampling and oversampling techniques can both deal with this problem, but the last is generally preferred because it preserves all the available data without discarding valuable information (*Mujahid et al., 2024*). In previous studies on classification of theropod teeth the importance of oversampling methods has often been underestimated. To our knowledge, *Wills, Underwood & Barrett (2020)* and *Ballell, Benton & Rayfield (2022)* are currently the only

studies that have used oversampling methods to address data imbalance. However, *Ballell, Benton & Rayfield (2022)* did not specify the method used, whereas *Wills, Underwood & Barrett (2020)* used the Synthetic Minority Oversampling Technique (SMOTE; *Chawla et al. (2002)*).

Machine learning models have been shown to perform well at processing large, complex datasets and can incorporate various variables to improve classification metrics such as accuracy. These models can handle class imbalances effectively, often outperforming traditional morphometric methods like PCA and LDA (*Wills, Underwood & Barrett, 2020*). Previous studies (*e.g.*, *Wills, Underwood & Barrett, 2020*; *Wills, Underwood & Barrett, 2023*; *Hendrickx et al., 2023*) have demonstrated the potential of combining ML techniques with traditional approaches, such as discriminant and cladistic analyzes, to classify isolated theropod teeth. ML methods have also been applied to predict dietary habits across dinosaur groups based on correlations between dental morphology and function (*Millar, 2019*; *Ballell, Benton & Rayfield, 2022*). However, despite these advances, the application of ML for theropod tooth classification remains in its early stages, requiring further development to improve accuracy and robustness (*Hendrickx et al., 2023*; *Wills, Underwood & Barrett, 2023*; *Yu et al., 2024*).

The issue of unbalanced data is particularly crucial in paleontological research (*Yu et al., 2024*). Traditional classifiers, such as DFA, often struggle with minority classes (*Xue & Hall, 2015*), leading to poor recognition of taxa that are less common in the fossil record. However, there are methodologies designed to handle data imbalances, such as those using SMOTE.

*Yu et al. (2024)* presents the first review on the application of ML techniques in Paleontology, where macrofossil classification studies are relatively limited compared to those focused on microfossils, and the range of the studied taxa is similarly restricted. However, they propose that the combination of large datasets on living organisms and a lack of trained experts highlights the strong potential for AI-powered automated workflows.

*Wills, Underwood & Barrett (2020)* utilized two published datasets, comprising 886 and 3,020 theropod teeth from 14 and 17 taxa, respectively, and analyzed five morphometric variables. They compared several classification methods, including linear discriminant analysis(LDA), logistic regression (LR), mixture discriminant analysis (MDA), naïve Bayes(NB), random forests (RF), and rule-based decision trees (C5.0), using ten-fold cross-validation to evaluate performance. To address missing data and class imbalance, the study also generated synthetic data using the SMOTE. SMOTE oversampling presented sub-optimal results and negatively affected the classification in methods used in the study. However, the study did not explore more advanced SMOTE-based oversampling techniques, such as K-Means SMOTE (KMeansSMOTE) or Support Vector Machine SMOTE (SVMSMOTE). These modified SMOTE methods could potentially improve results in imbalanced datasets. Additionally, the authors log-scaled the variables rather than more comprehensive standardization methods, which could have enhanced the effectiveness of some algorithms, particularly those sensitive to standardization of the variables.

*Wills, Underwood & Barrett (2023)* examined the performance of various machine learning techniques for classification of isolated theropod teeth. The models were built and trained using five morphometric variables derived from a combination of several published datasets, which covered a wide range of theropod genera from diverse geographic regions and time intervals. However, the datasets exhibited a notable bias toward teeth from North American Late Cretaceous genera, potentially leading to data imbalance that could affect model generalizability. The study applied three machine learning techniques namely, mixture discriminant analysis (MDA), random forests (RFs), and the C5.0 rule-based decision tree, and combined their results into an ensemble classifier to improve accuracy. Despite this, and following the methodology presented in *Wills, Underwood & Barrett (2020)*, no standardization or oversampling techniques were employed. The authors only log-scaled the variables, which may limit the model's robustness, particularly in addressing class imbalance issues.

*Hendrickx et al. (2023)* used a combination of discriminant function analysis (DFA), cladistic analysis, and various machine learning techniques to classify theropod teeth based on twelve crown and denticle-based measurement variables. The study utilized DFA with all variables log-transformed, while RFs, mixture discriminant analysis (MDA), and the C5.0 rule-based decision tree were applied using morphometric variables that were log-transformed, scaled, and centered. The dataset was split into training (80%) and testing (20%) subsets, preserving the class distribution of the original data, and evaluated using five-fold cross-validation. However, and following the methodology presented in *Wills, Underwood & Barrett (2020)*, the study did not use standardization methods or oversampling techniques, only log-scaling the variables, which may limit the model's ability to handle imbalanced data issues.

*Ballell, Benton & Rayfield (2022)* explored the use of nine different machine learning models, including neural networks and naïve Bayes (NB), to predict the diet of early dinosaurs. These models were applied to biomechanical and morphological datasets, which categorized dinosaurs into three dietary classes (carnivores, herbivores, and omnivores). To address class imbalance, an oversampling technique was used during the cross-validation process, although the exact method was not clearly defined in the study. The neural network model achieved the best performance on the morphological dataset, while NB outperformed other models in classifying biomechanical data. This study highlights the versatility of machine learning techniques in predicting ecological traits, such as diet, as opposed to their more common use in taxonomic classification.

*Millar (2019)* used machine learning techniques to predict the hunting behaviors of dinosaurs based on their physical characteristics, specifically distinguishing carnivorous dinosaurs as either scavengers or active hunters. The variables included in the model were: teeth length, estimated bite force, body weight or size, primary prey, estimated daily caloric needs, body length, maximum speed of both predator and prey, and sensory abilities (eyesight and sense of smell). The study applied several machine learning algorithms, including K-nearest neighbor (K-NN), LR, support vector machine (SVM), latent Dirichlet allocation, NB, and decision tree (DT). Of these, latent Dirichlet allocation, DT, and SVM achieved the highest accuracy in classifying dinosaur hunting habits. The best performance
of these models may be related to their ability to uncover complex patterns in the data and to create robust decision boundaries.

Machine learning and geometric morphometric approaches have become increasingly important for taxonomic identification of isolated theropod teeth, with significant advancements in dataset scale (*e.g.,* increasing size, diversity, and richness of dataset) and coverage over time. These methods are particularly valuable for addressing dataset imbalances inherent in paleontological studies. Recent studies, such as one focused on the Cenomanian Kem Kem Group of Morocco, used machine learning, cladistic, and discriminant analyses to classify isolated theropod teeth. This research highlighted how combining methods and using more diverse datasets for training can improve the reliability of classifications, especially because dataset imbalances strongly affect traditional techniques. Machine learning was particularly effective in distinguishing theropod taxa, even when sample sizes were uneven, helping to enhance the overall dataset's utility (*Wills, Underwood & Barrett, 2020*; *Hendrickx et al., 2023*). By using geometric morphometric techniques and incorporating larger, more diverse datasets, researchers have progressively increased coverage and improved our understanding of theropod dental diversity. These advancements demonstrate a significant shift toward minimizing the impact of dataset imbalance on paleontological inferences.

Data standardization is especially important when using methods like PCA, which is commonly used in paleontology, since differences in scale can disproportionately affect the resulting components or classification boundaries. This happens because PCA is based on variance that depends on the units of the data. Therefore, changing the units of one or more variables affects the principal components derived from the covariance matrix. If variables are measured on different scales, the results in principal components do not accurately reflect the true relationships between variables (*Jolliffe & Cadima, 2016*). Data standardization is also important for many machine learning algorithms, as it addresses the sensitivity of certain models to feature values. Without standardization, the model might incorrectly identify relationships in the data, leading to inaccurate predictions and poor performance. This pre-processing step ensures that the model focuses on true patterns rather than being skewed by variations in scale between features, thereby improving the overall effectiveness of the model in solving the intended task (*Izonin et al., 2022*). In *Jones & Close (2024)* the importance of standardization in paleontological datasets is mentioned and proven. They used a coverage-standardized disparity algorithm, where the standardization is achieved by calculating morphological variance or disparity while controlling for the unequal presence of samples across different time intervals.

Given the challenges of class imbalance and differences in variable scales we hypothesized that combining oversampling and standardization techniques would improve model robustness and classification reliability. Previous studies have not systematically tested multiple standardization or oversampling methods. To address this gap, we compare the performance of six machine learning models using different combinations of three standardization techniques and four oversampling methods. Our goal is to identify the most suitable approaches for taxonomic classification of isolated theropod teeth and for distinguish specimens from different positions along the tooth row and present best

practices for more reliable predictions. Although considerable progress has been made in the classification of dinosaur teeth, no study has made trained models freely available for application to new data, limiting researchers' ability to classify new findings using comparable models despite many studies have made their datasets available (*e.g.*, *Hendrickx et al., 2019*; *Hendrickx et al., 2023*). A key contribution of this study is the availability of trained models and standardization techniques that may be used on other datasets.

## METHODS

### Data processing and preparation for the modelling phase

The pre-processing phase is divided into three different stages: the first corresponds to data cleaning, which includes selection of variables and taxa, the second corresponds to the oversampling of the data, and the third corresponds to the standardization of the data.

The theropod teeth measurements were obtained from *Hendrickx et al. (2023)* (Fig. 1), which is an updated dataset with morphometric variables on a large set of theropod teeth compiled from previous works (*e.g.*, *Smith, Vann & Dodson, 2005*; *Hendrickx, Mateus & Araújo, 2015*; *Hendrickx, Tschopp & Ezcurra, 2020*). The variables initially evaluated were: (1) Crown Base Length (CBL); (2) Crown Base Width (CBW); (3) Crown Height (CH); (4) Apical Length (AL); (5) Crown Base Ratio (CBR); (6) Crown Height Ratio (CHR); (7) Mid-crown Length (MCL); (8) Mid-crown Width (MCW); (9) Mid-crown Ratio (MCR); (10) Mesiobasal Denticle Extension (MDE); (11) Mesial Serrated Carina Length (MSL); (12) Crown Angle (CA); (13) Mesioapical Denticle Density (MA); (14) Mesiocentral Denticle Density (MC); (15) Mesiobasal Denticle Density (MB); (16) Distoapical Denticle Density (DA); (17) Distocentral Denticle Density (DC); (18) Distobasal Denticle Density (DB), Distal Denticle Length (DDL); (19) Average Mesial Denticle Density (MAVG); (20) Average Distal Denticle Density (DAVG); and (21) Denticle Size Density Index (DSDI). The isolated theropod teeth were found mainly in the United States (U.S.- 435 observations), Canada (300 observations), Argentina (99 observations) and China (89 observations) (Table 1). The ontogenetic state of the individuals is mostly uncertain (Table 2), with most specimens belonging to adult and immature individuals in different ontogenetic stages. The geological epoch of the deposits where the teeth were collected was variate, with the most common epochs belonging to Late Cretaceous (724 observations), "Middle" Cretaceous (238 observations), and Late Jurassic (205 observations) (Table 3). For more information about the methodology for the measurement of the variables and information on the phylogenetic framework of the isolated theropod teeth used, see *Hendrickx, Mateus & Araújo (2015)*.

Data pre-processing began by removing variables that contained more than 15% missing values from the total number of isolated teeth, this resulted in the selection of the following variables: DC, DDL, CBW, CBL, CBR, CHR and CH (Fig. 2). The inclusion of both raw variables (CBW, CBL, CH) and their ratio-derived counterparts (CBR and CHR) in the analysis was chosen since most machine learning methods, such as gradient boosting trees and random forests are less sensitive to multicollinearity as they prioritize variable splits rather than linear dependencies. By keeping both types of variables, we increase

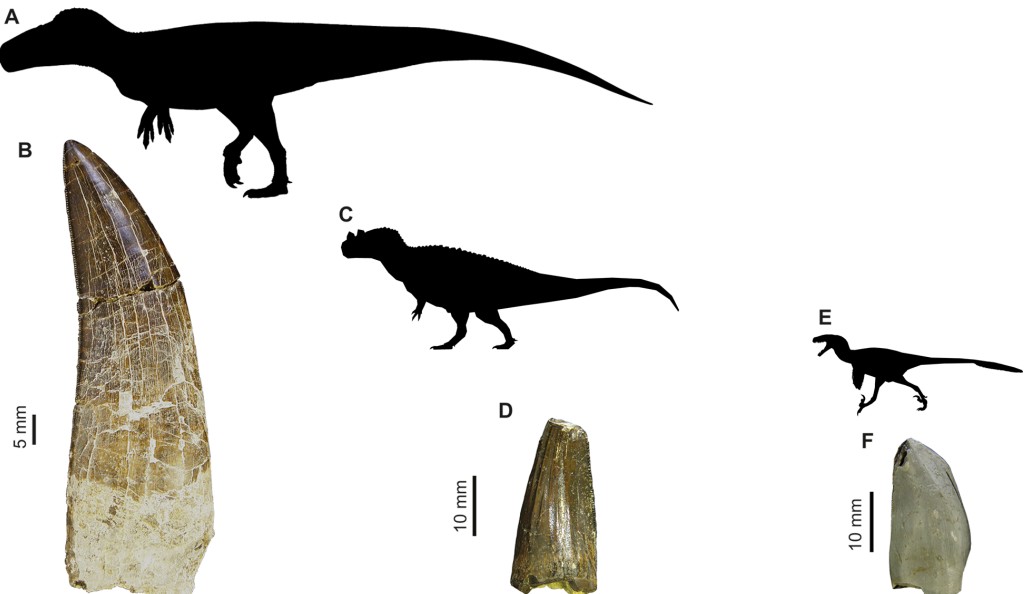

**Figure 1** Images of isolated theropod teeth from the Upper Jurassic of Portugal housed in the collection of Sociedade de História Natural (Torres Vedras, Portugal), published in *Malafaia et al. (2017)*. (A) Black silhouette of the *Torvosaurus tanneri* by Jagged Fang Designs, CC0 1.0 Universal Public Domain Dedication. (B) Shed tooth (SHN.401) in labial view, classified as *Torvosaurus*. (C) *Ceratosaurus nasicornis* by Tasman Dixon, CC0 1.0 Universal Public Domain Dedication. (D) Mesial shed tooth (SHN.457) in lingual view, classified as *Ceratosaurus* (E) Black silhouette of the *Dromaeosaurus albertensis* by Pranav Iyer, CC0 1.0 Universal Public Domain Dedication. (F) Possibly mesial shed tooth (SHN.278) in labial view classified as Coelurosauria, possibly related to *Richardoestesia*.

the possibilities of the models identifying subtle and crucial patterns in the dataset that might otherwise be missed if ratios or raw variables were excluded. After data cleaning, two datasets were created. The first for classification at the genus level, and the second for classification at a higher taxonomic level. The second dataset allowed for the inclusion of groups with fewer data cases that were excluded from the genus-level dataset to avoid class imbalance.

To maintain dataset quality, taxa within each dataset that contained observations equal to or fewer than the number of selected variables were removed, following the methodology of *Wills, Underwood & Barrett (2020)*. After this data processing, the same method of removal was applied to the previously excluded taxonomic groups, but removing the information for the tooth position, which was done to keep as much information as possible. After these steps, the higher taxonomic-level dataset contained 1,119 isolated theropod teeth belonging to twenty-three combinations of taxa and tooth position, including Abelisauridae lateral, Abelisauridae mesial, Allosauridae lateral, Allosauridae mesial, early-branching Theropoda, Carcharodontosauridae lateral, Carcharodontosauridae mesial, Dromaeosauridae lateral, Dromaeosauridae mesial, Metriacanthosauridae lateral, Neovenatoridae lateral, Noasauridae lateral, Non-abelisauroid Ceratosauria lateral,

**Table 1** Number of isolated theropod teeth by country, in the dataset used, sorted in decreasing order (only countries with more than 10 specimens).

| Country | Number of isolated theropod teeth |
| --- | --- |
| U.S. | 435 |
| Canada | 300 |
| Argentina | 99 |
| China | 89 |
| Madagascar | 78 |
| Morocco | 70 |
| U.K. | 57 |
| France | 50 |
| Mongolia | 50 |
| USA | 31 |
| Niger | 24 |
| Portugal | 23 |
| Australia | 17 |
| Japan | 15 |
| Germany | 14 |
| India | 11 |

**Table 2** Ontogenetic states for the isolated teeth from the dataset used, sorted by decreasing number of specimens (only ontogenetic state with more than 10 specimens).

| Ontogenetic state | Number of isolated theropod teeth |
| --- | --- |
| Adult | 244 |
| Adult? | 122 |
| Adult; Stage 4 | 80 |
| Adult or near adult | 53 |
| Immature? | 34 |
| Immature (juvenile) | 29 |
| Fairly mature | 26 |
| Subadult | 25 |
| Juvenile/immature | 25 |
| Immature | 22 |
| Old adult (28 yo) | 22 |
| Immature (young) | 16 |
| Adult or subadult | 13 |
| Immature (young); Stage 1 | 11 |

Non-abelisauroid Ceratosauria mesial, Non-averostran Neotheropoda lateral, Non-spinosaurid Megalosauroidea lateral, Non-tyrannosaurid Tyrannosauroidea lateral, Non-tyrannosaurid Tyrannosauroidea mesial, Spinosauridae, Therizinosauria, Troodontidae lateral, Tyrannosauridae lateral, Tyrannosauridae mesial; and the genus-level dataset contained 925 isolated theropod teeth from thirty-seven combinations of genera and tooth position, including *Acrocanthosaurus* lateral, *Albertosaurus* lateral, *Alioramus* lateral,

**Table 3  Geological epoch ordered by decreasing number of specimens.**

| Geological epoch | Number of isolated theropod teeth |
| --- | --- |
| Late Cretaceous | 724 |
| "Middle" Cretaceous | 238 |
| Late Jurassic | 205 |
| Early Cretaceous | 81 |
| Late Triassic | 57 |
| Middle Jurassic | 55 |
| Early Jurassic | 9 |
| Late Jurassic - Early Cretaceous | 2 |

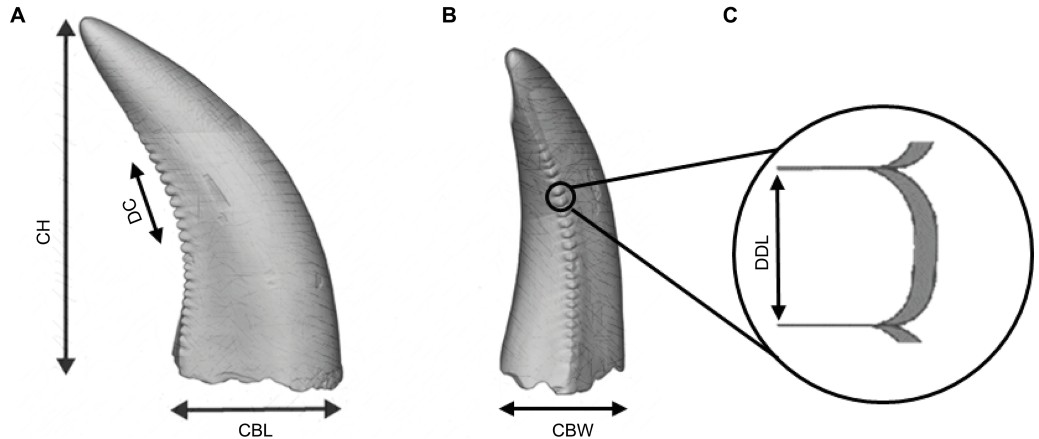

**Figure 2  Illustration of an isolated theropod teeth, adapted from Figure 2 of *Wills, Underwood & Barrett (2023)*, licensed under the CC BY 4.0. Modifications include, color adjustments, measurements modifications, removal of basal view and inclusion of denticle section.** The illustration includes the isolated teeth with the morphometric variables used by the classifiers. (A) Lingual view of the tooth. (B) Distal view of the tooth. (C) Zoom figure of the denticles.

*Allosaurus* lateral, *Allosaurus* mesial, *Atrociraptor* lateral, *Australovenator*, *Bambiraptor* lateral, *Baryonyx* lateral, *Carcharodontosaurus* lateral, *Ceratosaurus* lateral, *Ceratosaurus* mesial, *Coelophysis* lateral, *Daspletosaurus* lateral, *Deinonychus* lateral, *Dilong* lateral, *Dromaeosaurus* lateral, *Eoraptor*, *Fukuiraptor* lateral, *Gorgosaurus* lateral, *Majungasaurus* lateral, *Majungasaurus* mesial, *Masiakasaurus* lateral, *Monolophosaurus*, *Pectinodon* lateral, *Raptorex* lateral, *Richardoestesia* lateral, *Saurornitholestes* lateral, *Sinraptor*, *Suchomimus* lateral, *Torvosaurus* lateral, *Troodon* lateral, *Tyrannosaurus* lateral, *Tyrannosaurus* mesial, *Velociraptor* lateral, *Zapsalis* lateral.

To obtain more reliable estimate, k-fold cross-validation was used to generate the training and testing datasets, as has been done in other works (*Wills, Underwood & Barrett, 2020*; *Hendrickx et al., 2023*). The k-fold cross-validation divides the dataset into k subsets (folds), with each fold used as a testing set while the remaining k-1 folds are used for training. Following the methodology used by *Hendrickx et al. (2023)*, a value of

five was chosen for k. Stratified k-fold cross-validation was used instead of standard k-fold cross-validation to ensure that each fold preserves the class distribution within the dataset, preventing loss of information from underrepresented classes. After splitting the data using cross-validation, oversampling and standardization was applied to the training dataset, and the same standardization parameters were subsequently applied to the testing dataset. The different methods of oversampling were applied to avoid possible model bias for more represented taxa. The following four oversampling techniques used in previous studies (*e.g.*, *Nguyen, Cooper & Kamei, 2011*; *Douzas, Bacao & Last, 2018*; *Wills, Underwood & Barrett, 2020*) were evaluated: (1) Random Oversampling (RandomOverSampler), which repeats values (*Fernández et al., 2018*) and is a non-heuristic algorithm; (2) SMOTE, which generates synthetic samples for the minority class by interpolating between existing minority samples (*Chawla et al., 2002*); (3) Borderline Synthetic Minority Oversampling Technique (BorderlineSMOTE), which is a variant of SMOTE, but focuses on generating synthetic samples near the borderline of the classes (*Han, Wang & Mao, 2005*); and (4) K-Means Synthetic Minority Oversampling Technique (KMeansSMOTE), which uses K-Means clustering to generate synthetic samples by clustering the minority class and then applying SMOTE within each cluster (*Douzas, Bacao & Last, 2018*). We then compared three scaling methods, which were chosen following *Ahsan et al. (2021)* that analyzes the effect of different standardization techniques on diverse ML model performances. Firstly, we used the standard scaler (StandardScaler), which transforms the data so that the mean is 0 and the standard deviation (SD) is 1 (*Sola & Sevilla, 1997*). It uses the $Z$-score normalization, where the standardized value is given by:

$$z = \frac{(x - \mu)}{\sigma} \tag{1}$$

where $x$ is the original value, $\mu$ the mean value of the variable, and $\sigma$ is the SD of the variable.

Secondly, we used the robust scaler (RobustScaler), which scales the data according to the inter-quartile range (IQR) (*Aggarwal, 2017*), where the standardized value is given by:

$$x' = \frac{x - x_{median}}{Q_3 - Q_1} \tag{2}$$

where $x_{median}$ is the median value of the variable, and $Q1$ and $Q3$ refer to the first and third quartiles, respectively.

Finally, we used the Quantile Transformer (QuantileTransformer), which transforms the data to follow a normal distribution, with values between 0 and 1 (*Pedregosa et al., 2011*), where the standardized value is obtained by first getting the value of:

$$x' = \Phi^{-1}\left(\frac{\text{rank}(x)}{N + 1}\right) \tag{3}$$

where $\Phi^{-1}$ is the inverse of the cumulative distribution function of the standard normal distribution, $rank(x)$ is the rank of the value $x$ in the sorted data, and $N$ is the total number of data points.

## Data modeling

Using the training and testing datasets, we evaluated the performance of the following six machine learning multi-class classification models: (1) RF, which was previously used in theropods teeth classification (*Wills, Underwood & Barrett, 2020*; *Wills, Underwood & Barrett, 2023*; *Hendrickx et al., 2023*); (2) gradient boosting (GB), which from our knowledge has not yet been used on theropod studies, but has been applied in different dentistry problems (*Carrillo-Perez et al., 2021*); (3) K-NN and (4) SVM, which were used to distinguish different types of dinosaur hunters (*Millar, 2019*); (5) NN, which was previously used to infer dinosaur diets based on dental morphology (*Ballell, Benton & Rayfield, 2022*); and (6) discriminant analysis, which has been commonly used on theropods teeth classification (*e.g.*, *Wills, Underwood & Barrett, 2020*; *Wills, Underwood & Barrett, 2023*; *Hendrickx et al., 2023*). RF is an ensemble method that uses multiple decision trees (*Breiman, 2001*). This model uses Gini index to measure the quality of a split. For example, if a target is a classification outcome taking on values 0, 1, ..., $k-1$, for node m, the proportion of class $k$ observations in node m is given by:

$$p_{m,k} = \frac{1}{n_m} \sum_{y \in Q_m} I(y = k) \tag{4}$$

The measure of the impurity is given by 'Gini' *Yuan, Wu & Zhang (2021)* which is obtained by:

$$H(Q_m) = \sum_k p_{mk}(1 - p_{mk}) \tag{5}$$

where $Q_m$ represents the data node at node $m$, and $n_m$ is the number of samples. The minimum number of samples required to split an internal node is 2. The minimum number of samples required to be at a leaf node is 1 and the maximum depth of the decision trees was 22. GB is also an ensemble technique that builds trees sequentially, where each tree corrects errors of the previous ones (*Friedman, 2001*). We used a learning rate of 0.1 and the log-loss function (*Murphy, 2012*), is given by:

$$\text{Log Loss} = -\big[y \cdot \log(p) + (1 - y) \cdot \log(1 - p)\big] \tag{6}$$

where $y$ is the true label, $p$ is the predicted probability for a given class. The number of boosting stages to perform was 100, the fraction of samples used for fitting the individual base learners was 1, and the function to measure the quality of a split was the mean squared error (MSE) with improvement score by *Friedman (2001)*, which is given by:

$$\text{Friedman MSE} = \frac{1}{N} \sum_{i=1}^{N} \big((y_i - \hat{y}_i)^2\big) - \lambda \cdot \frac{1}{N} \sum_{i=1}^{N} \hat{y}_i^2 \tag{7}$$

where $N$ is the number of samples, $y_i$ is the actual target value for the $i$th sample, and $\lambda$ is a regularization parameter that controls the trade-off between variance and bias. K-NN method classifies a sample based on the majority class of its K nearest neighbors (*Cover & Hart, 1967*). We used $K = 5$, and all points in each neighborhood are weighted equally,
following a uniform distribution. The metric used for the distance computation was the Minkowsk distance (*Jain, Duin & Mao, 2000*), which is obtained by:

$$\|u-v\|_2 = \left(\sum |u_i - v_i|^p\right)^{1/p} \tag{8}$$

where $u$ and $v$ are input arrays, $p$ is the order of the norm of the difference $|u-v|$. NN method uses layers of neurons to learn (*Goodfellow, Bengio & Courville, 2016*). The models' hyper-parameters were determined through preliminary runs. The model was trained for 50 epochs of training, the number of training examples used in one forward and backward pass through the network (batch size) was 32. The network architecture comprised three fully connected layers: a first layer containing 128 neurons, followed by a second layer with 64 neurons, and a third layer with 32 neurons, all these layers with a Rectified Linear Unit (ReLU) activation function. The output layer was a Softmax layer, where the output size corresponded to the number of target classes in the dataset. The ReLU activation function (*Ying et al., 2019*) value is obtained by:

$$\text{ReLU}(x) = \max(0, x) \tag{9}$$

The optimizer used was the Adaptive Moment Estimation (Adam) *Kingma & Ba (2014)* which uses the following initialize moment vectors: $m_{(0)} = 0$ and $v_{(0)} = 0$, where $m_t$ and $v_{(t)}$ are estimates of the first moment (mean) and the second moment (uncentered variance) of the gradients respectively. The update biased first moment estimate is given by:

$$m_t = \beta_1 \cdot m_{t-1} + (1 - \beta_1) \cdot g_t \tag{10}$$

and the update biased second moment estimate is given by:

$$v_t = \beta_2 \cdot v_{t-1} + (1 - \beta_2) \cdot g_t^2 \tag{11}$$

where $g_t$ is the gradient at step $t$, $\beta_1$ and $\beta_2$ are the decay rates for the moment estimates. To compute the bias-corrected first moment estimate the following formula is used:

$$\hat{m}_t = \frac{m_t}{1 - \beta_1^t} \tag{12}$$

To compute the bias-corrected second moment estimate the following formula:

$$\hat{v}_t = \frac{v_t}{1 - \beta_2^t} \tag{13}$$

The parameter update rule is given by:

$$\theta_t = \theta_{t-1} - \alpha \cdot \frac{\hat{m}_t}{\sqrt{\hat{v}_t} + \epsilon} \tag{14}$$

where $\theta_t$ represents the parameters being optimized, $\epsilon$ is a small constant to prevent division by zero and $\alpha$ is the learning rate.

The loss function used was the Sparse Categorical Cross-Entropy (*Abadi et al., 2016*) and its values is given by:

$$\text{Loss} = -\frac{1}{N} \sum_{i=1}^{N} \log(p(y_i)) \tag{15}$$

where $p(y_i)$ is the predicted probability of the true class $y_i$, $y_i$ is the true class label for the $i$th sample and $N$ is the number of samples. The Softmax *Bridle (1989)* value is given by:

$$\sigma(z_i) = \frac{e^{z_i}}{\sum_{j=1}^{K} e^{z_j}} \tag{16}$$

where $z$ is a vector with the raw prediction scores, $K$ is the number of classes, $e^{z_i}$ is the exponential of the score for class $i$, $\sum_{j=1}^{K} e^{z_j}$ is the sum of exponentials of all the scores across all classes, and $\sigma(z_i)$ represents the probability of the input belonging to class $i$. The model learns after each epoch of training while minimizing the loss value. Quadratic discriminant analysis (QDA) model's class-specific covariance matrices allowing the formation of quadratic decision boundaries, where the class proportions are inferred from the training data and by fits class conditional densities to the data and using Bayes' rule (*Friedman, 1989*), given by:

$$P(C_k|x) = \frac{P(x|C_k)P(C_k)}{P(x)} \tag{17}$$

where $P(C_k|x)$ is the posterior probability of class $k$ given the feature vector $x$, $P(x|C_k)$ is the likelihood of observing $x$ given that it belongs to class $k$, $P(C_k)$ is the prior probability of class $k$, $P(x)$ is the marginal likelihood of $x$. The absolute threshold for a singular value to be considered significant is 1.0e−4. The final classification decision is based on the equation:

$$\hat{C}(x) = \arg\ \max\left(\frac{1}{2}\log|\Sigma_k| - \frac{1}{2}(x-\mu_k)^T\Sigma_k^{-1}(x-\mu_k) + \log P(C_k)\right) \tag{18}$$

where $\mu_k$ is the mean vector for class $k$, $\Sigma_k$ is the covariance matrix for class $k$ and $x$ is the feature vector. A regularization parameter ($\lambda$) was included with a value of 0.1. This parameter regularizes the per-class covariance estimates by transforming the scaling attribute of a given class ($S^2$) as:

$$S^2 = (1-\lambda)\Sigma + \lambda I_n \tag{19}$$

where $\Sigma$ is the estimated covariance matrix of the data, $n$ corresponds to the number of features and $I_n$ to the identity matrix. Finally, SVM method finds the optimal hyperplane to separate different classes in the feature space (*Cortes & Vapnik, 1995*). The kernel type used in the algorithm was the radial basis function kernel (rbf) (*Ding et al., 2021*), which is given by:

$$K(x_1, x_2) = \exp(-\gamma \cdot \|x_1 - x_2\|^2) \tag{20}$$

where $\gamma$ is the influence of each individual training sample on the decision boundary, $x_1$ and $x_2$ are two data points and $\|x_1 - x_2\|^2$ is the Euclidean distance between the two points. The degree of the polynomial kernel function is 3.

A total of 120 model combinations (classifier, oversampling and standardization methods) were trained and evaluated. Since we used two different classification tasks and used two different variable types (log-scaled and non-log-scaled), we had a total of 480 models trained, evaluated and compared.

## Model evaluation metrics

All models used a random number generator, and a fixed seed was used for reproducibility. The influence of the variables on the classification results was assessed using permutation feature importance. This technique evaluates the significance of each variable in a predictive model by randomly shuffling the values of a single feature, thereby breaking its relationship with the target. The resulting degradation in the model's performance indicates how much the model relies on that particular feature.

All models were compared using different metrics, including accuracy (correct classification rate), precision (positive predictive value), and F1-score. These metrics are given by:

$$\text{Accuracy} = \frac{TP + TN}{TP + TN + FP + FN} \tag{21}$$

$$\text{Precision} = \frac{TP}{TP + FP} \tag{22}$$

$$\text{Recall} = \frac{TP}{TP + FN} \tag{23}$$

$$\text{F1-Score} = \frac{2 \times \text{Precision} \times \text{Recall}}{\text{Precision} + \text{Recall}} \tag{24}$$

where $TP$ represents the correctly predicted positive cases, $TN$ represents the correctly predicted negative cases, $FP$ represents the cases incorrectly predicted as positive, and $FN$ represents the cases incorrectly predicted as negative.

## Comparison between log-scaled and non-log-scaled variables

Log-scaling is a data transformation technique used to address skewed distributions and reduce the impact of outliers. By compressing the range of values, particularly in data spanning multiple orders of magnitude, this transformation can reveal underlying patterns and potentially improve model performance (*Iglewicz & Hoaglin, 1993*). Log-scaling has been commonly applied in morphometric analyzes of isolated theropod teeth (*e.g.*, *Malafaia et al., 2017*; *Hendrickx et al., 2019*; *Wills, Underwood & Barrett, 2020*; *Wills, Underwood & Barrett, 2023*; *Hendrickx et al., 2023*). To assess the effect of this transformation, a comparison was made between models trained on non-log-scaled and log-scaled variables to evaluate its impact on model performance. This comparison was done by computing the difference between the mean metrics across all the testing folds using the non-log-scaled and log-scaled variables. Log-scaling was done using the natural logarithm.

## Implementation information

All data pre-processing, including variables and taxa selection, as well as the correlation analyzes, was performed with the R Software (*R Core Team, 2022*), version 4.2.1. All standardizations and oversampling methods were applied using Python (*Python*

**Table 4** **Summary of the different information on the tested isolated theropod teeth, including the measurements used for the classification model.** All measurements are in millimeters (mm).

| Inventory number | Probable position | Initial classification | CBL | CBW | CBR | CHR | DC | DDL | CH |
|---|---|---|---|---|---|---|---|---|---|
| MG 27782 | mesial | Tyrannosauridae | 3.69 | 2.79 | 0.76 | 0.51 | 30.0 | 1.18 | 7.28 |
| MG 27768 | lateral | Tyrannosauridae | 6.28 | 3.09 | 0.49 | 0.47 | 30.0 | 0.28 | 13.3 |
| MG 27805_D193 | lateral | *Ceratosaurus* | 4.78 | 2.14 | 0.45 | 0.61 | 27.5 | 0.52 | 7.76 |
| MG 27808_D118 | lateral | *Richardoestesia* | 2.43 | 1.01 | 0.41 | 0.42 | 41.2 | 0.18 | 5.78 |

*Software Foundation, 2024*), version 3.8.16, using scikit-learn (*Pedregosa et al., 2011*) and imbalanced-learn (*Lemaître, Nogueira & Aridas, 2017*) respectively. All metrics values and permutation results were obtained using scikit-learn (*Pedregosa et al., 2011*). All the models (except the NNs) were created using scikit-learn (*Pedregosa et al., 2011*), and the NN models were created using TensorFlow (*Abadi et al., 2016*). The trained models and scalers corresponding to the best models for each classification task can be found in Github. These trained models can be used directly to classify the probability of new isolated theropod teeth belonging to each class.

### Application of the methodology in the sample of isolated theropod teeth from Guimarota fossil site

The classification methods were also tested on a sample of unpublished theropod teeth (except MG 27808_D118 that was studied by *Zinke (1998)* based on morphological comparisons) collected during the last decades of the 20th century in the Kimmeridgian (Upper Jurassic) Guimarota fossil site (Leiria, Portugal). The initial classification here mentioned is based on the labels associated with the specimens, which probably correspond to preliminary identifications made by German researchers that first processed the fossil record from this locality (Table 4 and Fig. 3). These specimens are currently housed by Museu Geológico of Laboratório Nacional de Engenharia e Geologia (LNEG- Lisboa).

## RESULTS

The results for each combination of classification models, oversampling, and standardization methods were compared and analyzed. The complete set of results is available in the Supplemental Information. In this section, we discuss the accuracy of the models with the best and worst performance. All metric values presented here are averages derived from five-fold cross-validation, corresponding to the testing folds. We first present the classification results at the genus level, and then the higher taxonomic-level.

### Classification at genus level

The original dataset contained information of 1,371 isolated theropod teeth belonging to 93 genera. The dataset after data processing includes 909 isolated theropod teeth from thirty-six combinations of genera and tooth positions. There are two predominant genus combinations: *Saurornitholestes* lateral (133 observations) and *Tyrannosaurus* lateral (106 observations) (Fig. 4). After applying oversampling techniques, all training classes were balanced to contain the same number of observations each.

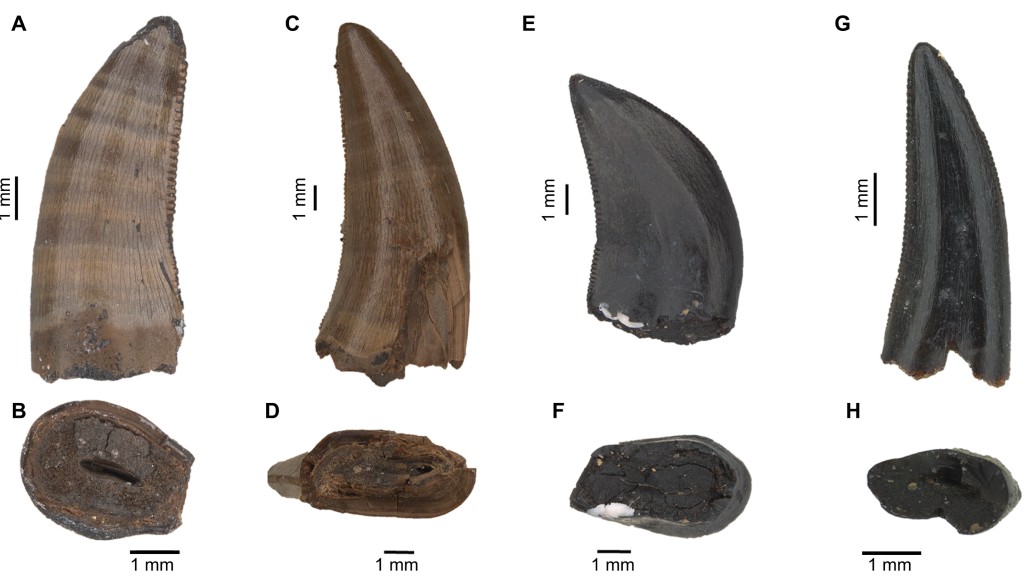

**Figure 3   Isolated theropod teeth used for testing the models with the best performance on new dataset.** (A) MG 27782 in labial view; (B) MG 27782 in basal view; (C) MG 27768 in labial view; (D) MG 27768 in basal view; (E) MG 27805_D193 in labial view. (F) MG27805_D193 in basal view; (G) MG 27808_D118 in labial view; (H) MG 27808_D118 in basal view.

The oversampling influence on the models varied between the models. While GB and SVM obtained lower evaluation metrics when oversampling methods were not applied, the K-NN, NN and QDA classifiers presented higher metric results when not using oversampling methods (see Fig. 5 for F1-score and the ShinyApp for the other metrics results). The model performance varied significantly across oversampling and scaling techniques. RF and GB were the best performing models, with consistent results between different oversampling and standardization techniques. GB benefited from RandomOversampler, K-NN and NN improved their performance by using standardization methods (*e.g.*, StandardScaler and RobustScaler), but they remained less effective when compared to RF and GB. QDA and SVM presented the worst overall metric results and an increase in the metric results when using standardization techniques, namely with StandardScaler and QuantileTransformer (see Fig. 5 for F1-score and the ShinyApp for the other metrics results).

For the non-log-scaled variables, RF and GB were the top-performing classifiers, consistently achieving high metric values across different oversampling and standardization approaches, with average performance metrics ranging from 67.3% to 75.6% on the test set and consistently 100% on the train set. The SVM and QDA models produced the worst results, with metric values ranging from 64.1% to 74.1% on the test and from 74.3% to 80.9% on the train set. Among the tested oversampling methods, BorderlineSMOTE and SMOTE achieved the highest metrics, with testing results ranging from 66.8% to 73.4% across different models and standardization approaches. In contrast, NoOversampling and KMeansSMOTE had the worst performance, with metric values ranging from 66.6% to

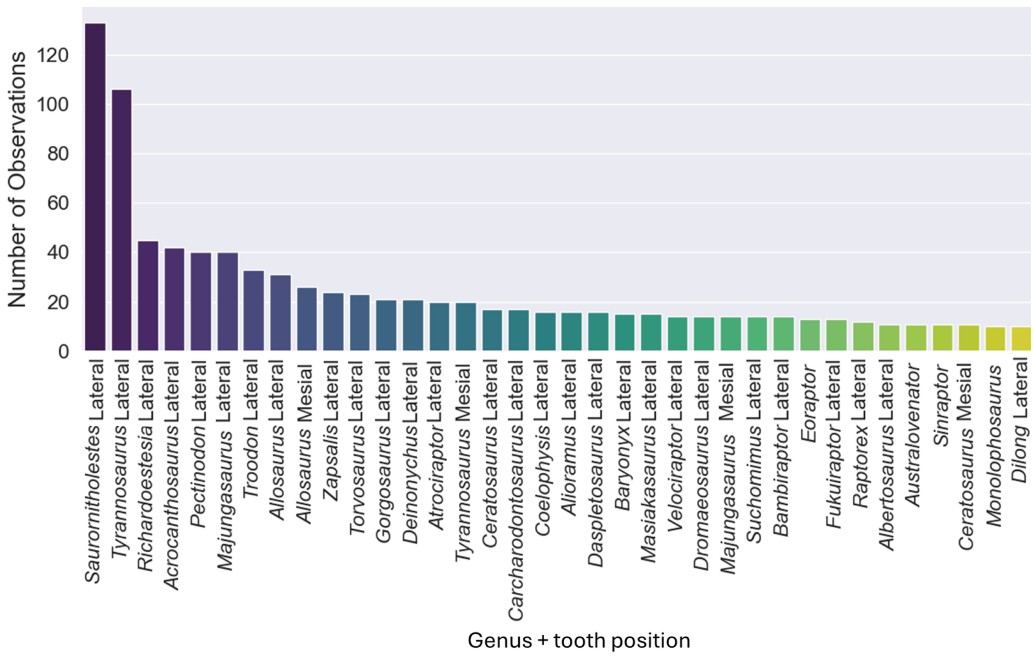

**Figure 4** Number of observations for each of the thirty-seven combinations of genera and tooth positions, showing the imbalance of the observations for each class.

73.7%. Regarding standardization techniques, StandardScaler and QuantileTransformer outperformed the other methods, with testing average metric values ranging from 67.8% to 73.7% across different models and oversampling approaches. In contrast, NoScaler and RobustScaler had the lowest performance, with metric values ranging from 64.8% to 73.5%.

For the log-scaled variables, RF and GB were the top-performing classifiers, with average metric values ranging from 66.9% to 74.7% across different oversampling and standardization approaches. In contrast, SVM and QDA had the worst performance, with metrics ranging from 62.4% to 74.2%. Among the oversampling methods, RandomOverSampler and SMOTE performed best, with metrics between 66.7% and 73.1%, while NoOversampling and KMeansSMOTE showed the lowest performance, with values between 65.5% and 73.7%. Regarding standardization techniques, RobustScaler and StandardScaler, as with the non-log-scaled variables, were the best, achieving metric values between 67.3%. and 73.5%. The worst-performing methods were NoScaler and QuantileTransformer, with metric values ranging from 63.9% to 72.6%.

The best-performing combination for both log-scaled and non-log-scaled variables was the RF model using either the RandomOverSampler oversampling using (1) no standardization, (2) StandardScaler standardization. It achieved F1-scores of 76.0% (SD = 1.3%) and 75.9% (SD = 1.9%) for non-log-scaled and log-scaled variables, respectively. Since the standardization method that presented better overall results was the StandardScaler, we will use the model combination of RF with StandardScaler and RandomOverSampler method for further analysis as the best model combination. The
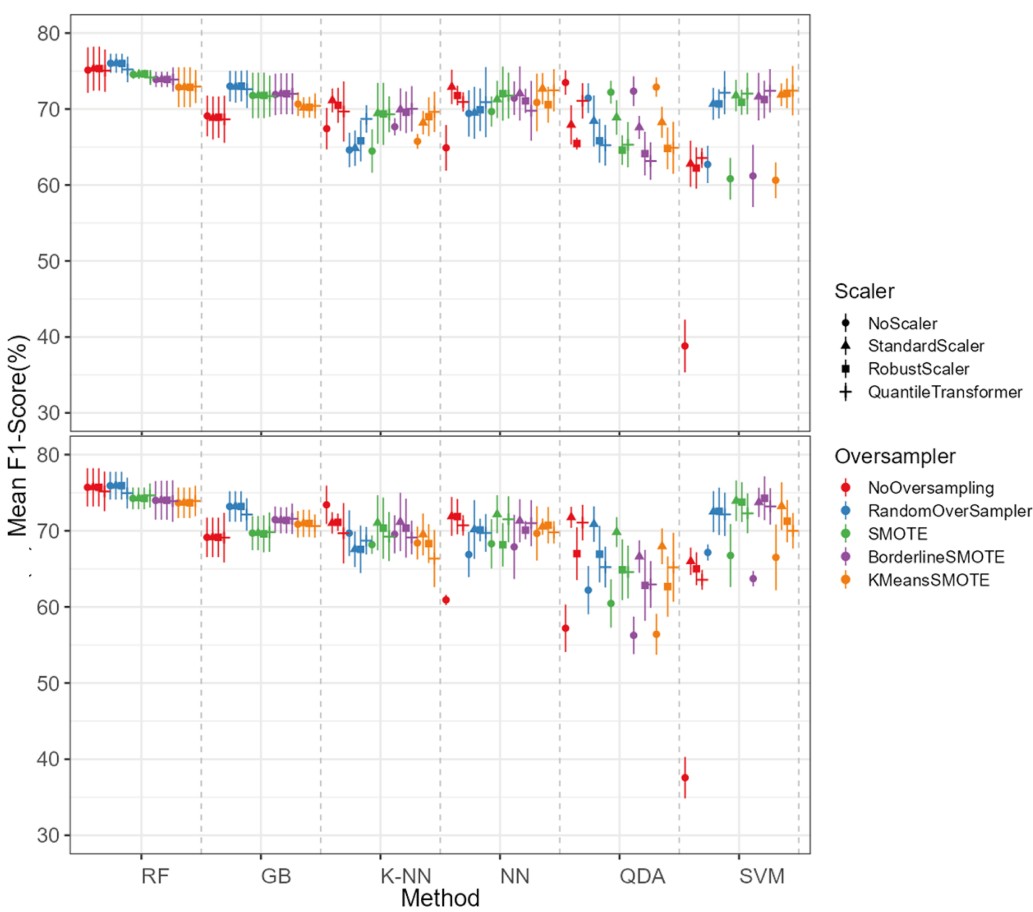

**Figure 5** **F1-score values for each combination of models, oversampling techniques, and scalers.** The top plot shows the results for models using non-log-scaled variables, while the bottom plot shows the results for models using log-scaled variables. The dotted line indicates the separation between different methods.

worst-performing combination was the SVM model not using both oversampling and standardization, with F1-scores of 37.6% (SD = 2.7%) and 38.8% (SD = 3.5%) for non-log-scaled and log-scaled variables, respectively. Log-scaled variables presented overall better results, having an averaged difference of 0.2% on the F1-scores when compared to the non-log-scaled variables.

The non-log-scaled model showed that the most common misclassifications were: (1) *Velociraptor* lateral teeth as *Saurornitholestes* lateral teeth, *Suchomimus* lateral teeth as *Baryonyx* lateral teeth, (2) *Tyrannosaurus* mesial teeth as *Tyrannosaurus* lateral teeth, (3) *Sinraptor* teeth as *Ceratosaurus* lateral teeth or as *Allosaurus* mesial, (4) *Monolophosaurus* as *Alioramus* lateral or as *Allosaurus* mesial, (5) *Masiakasaurus* lateral as *Saurornitholestes* lateral, (6) *Majungasaurus* mesial as *Allosaurus* mesial or as *Majungasaurus* lateral , (7) *Gorgosaurus* lateral as *Allosaurus* lateral, (8) *Fukuiraptor* lateral as *Alioramus* lateral or as *Monolophosaurus*, (9) *Dromaeosaurus* lateral as *Saurornitholestes* lateral, (10) *Daspletosaurus* lateral as *Acrocanthosaurus* lateral or as *Tyrannosaurus* lateral, (11) *Coelophysis* lateral as

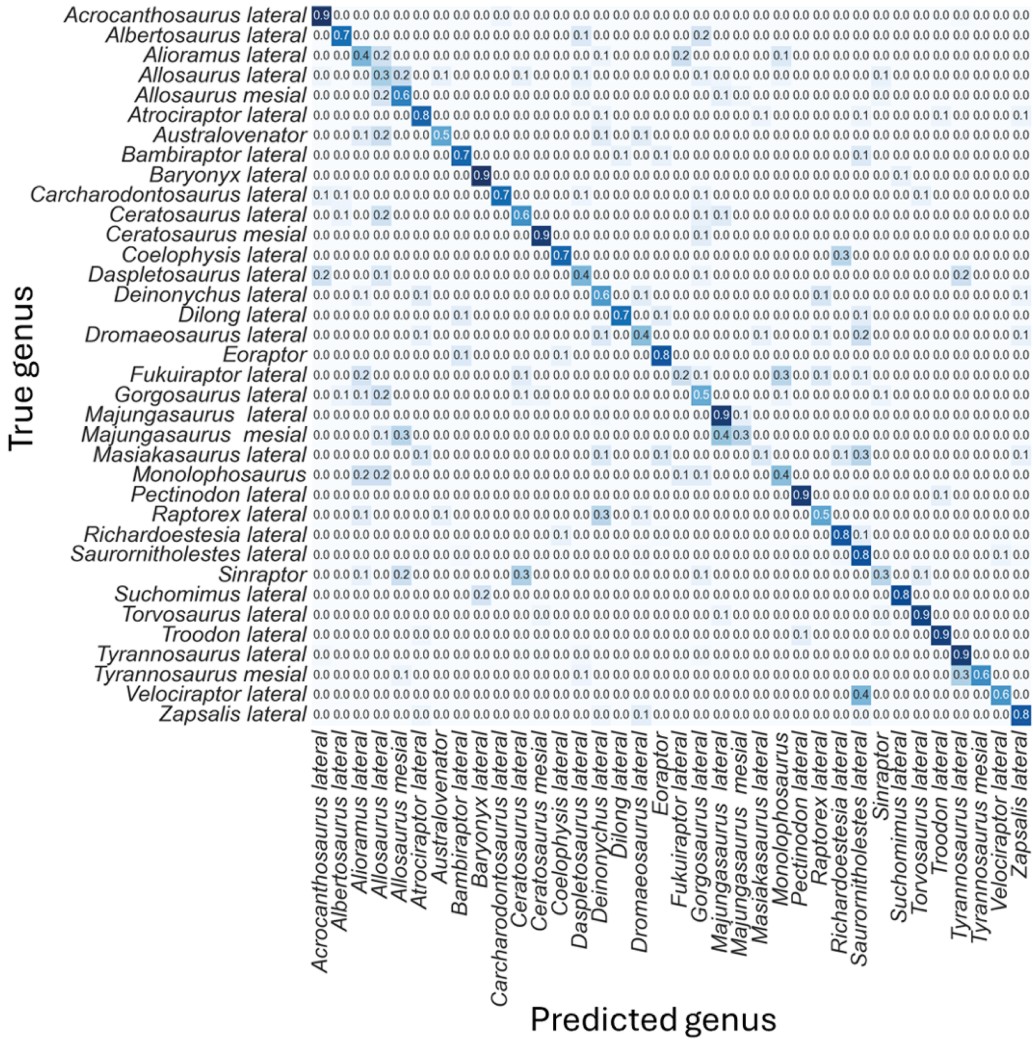

**Figure 6** Normalized confusion matrix for the mean proportion of predicted labels for the different classes.

*Richardoestesia* lateral, (12) *Ceratosaurus* lateral as *Allosaurus* lateral, (13) *Australovenator* as *Allosaurus* lateral, (14) *Allosaurus* lateral as *Allosaurus* mesial, (15) *Allosaurus* mesial as *Allosaurus* lateral, (16) *Alioramus* lateral as *Fukuiraptor* lateral or as *Allosaurus* lateral, (17) *Albertosaurus* lateral as *Gorgosaurus* lateral (Fig. 6).

Even-though the confusion matrix only shows the predicted class, the three most probable classes were evaluated in the cases were the prediction was incorrect for the whole dataset, resulting in a total of 98.5% correct predictions.

The most influential variable for the predictions was CBW, with a mean importance of 0.22 (Fig. 7), followed by DC, with a mean importance of 0.20 (Fig. 7).

The first branches of the decision trees show that the first criteria to do the classification was if DC is lower or equal to 0.675, the second decision was if (1) CBL was lower or equal to −0.608, (2) DC lower or equal to −0.424 (Fig. 8).
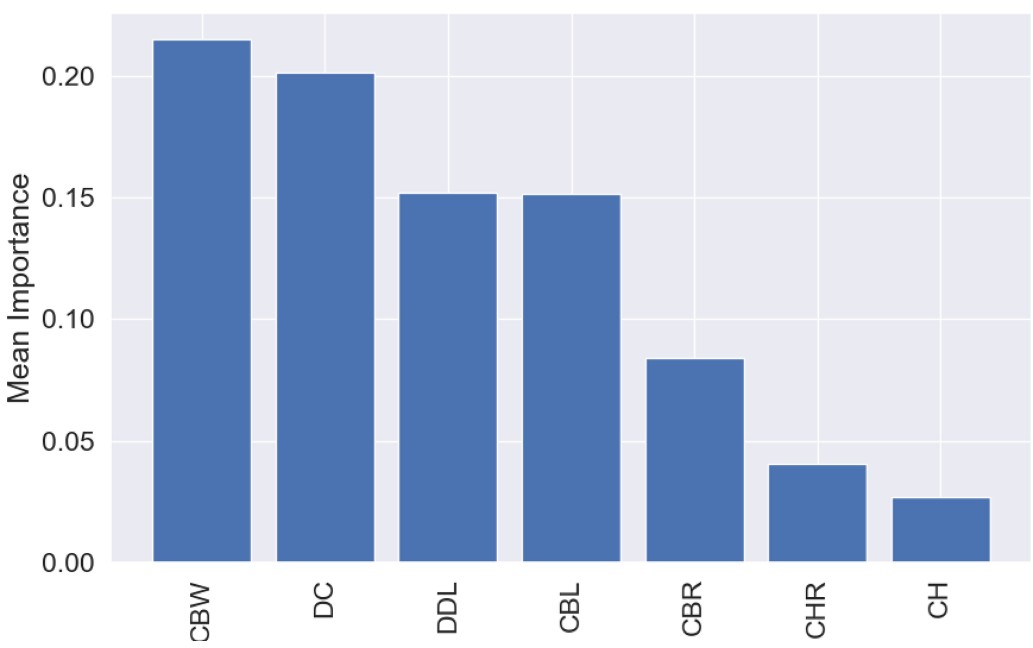

**Figure 7** Mean permutation importance of the different variables for the best classification model.

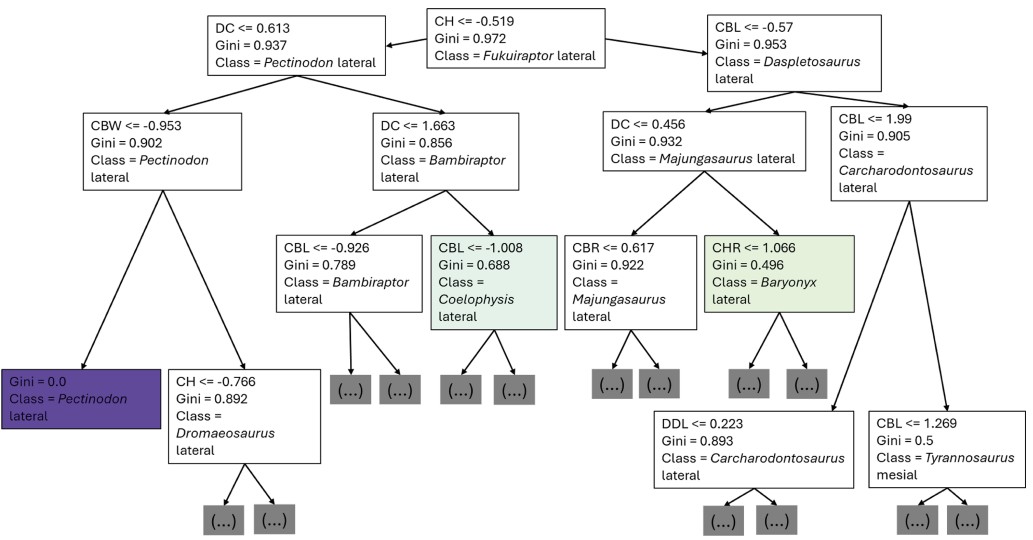

**Figure 8** First three branches of decisions of the first decision tree of the best model.

The best performing model was used to classify the set of isolated theropod teeth from Guimarota fossil site and all of the teeth were attributed to their corresponding initial classification in the three most probable classifications, except *Ceratosaurus* lateral (Table 5).

**Table 5   The three most probable classifications using the best performing model.**

| Inventory number | Most probable label | Second probable label | Third probable label |
|---|---|---|---|
| MG 27805_D193 | *Saurornitholestes* lateral | *Troodon* lateral | *Masiakasaurus* lateral |
| MG 27782 | *Dilong* lateral | *Trodon* lateral | *Atrociraptor* lateral |
| MG 27768 | *Saurornitholestes* lateral | *Masiakasaurus* lateral | *Zapsalis* lateral |
| MG 27808_D118 | *Saurornitholestes* lateral | *Richardoestesia* lateral | *Bambiraptor* lateral |

## Classification at higher taxonomic level

The dataset initially included 1,371 theropod teeth from 22 taxa, but after processing, it contained 1,119 isolated theropod teeth belonging to twenty-three combinations of taxa and tooth positions. Two groups were predominant: Tyrannosauridae lateral (175 observations) and Dromaeosauridae lateral (295 observations) (Fig. 9). While the best-performing combination for the non-log-scaled variables was the RF model using StandardScaler and RandomOverSampler technique, achieving F1-scores of 72.0%, the best performing model for log-scaled variables was the RF using QuantileTransformer and RandomOverSampler technique, achieving F1-scores of 72.1%. In contrast, the worst-performing combination for the non-log-scaled variables was the SVM model with KMeansSMOTE oversampling and no standardization, with F1-scores of 44.7%, and for the non-log-scaled variables it was the QDA using BorderlineSMOTE oversampling and no standardization, achieving values of 47.8%. As previously seen on the genus-level classification, the model performance varied significantly across oversampling and scaling techniques. RF and GB were again the best performing models, with consistent results between different oversampling and standardization techniques. GB benefited from RandomOversampler and both RF and GB benefited from QuantileTransformer. NN and SVM improved their performance by using standardization methods (*e.g.*, StandardScaler and RobustScaler), but they remained less effective when compared to RF and GB. QDA and SVM presented the worst metric results and an increase in the metric results when using standardization techniques, namely with StandardScaler. (see Fig. 10 for F1-score and the ShinyApp for the other metrics results). The overall difference for the F1-scores between the log-scaled and non-log-scaled datasets was −0.5%. We will use for further analysis the log-scaled RF model using the QuantileTransformer standardization and RandomOverSampler, since it was the combination that presented best metric results.

The log-scaled model showed that the most common misclassifications were: (1) Noasauridae lateral teeth as Dromaeosauridae lateral, (2) Tyrannosauroidea mesial as Tyrannosauroidea lateral, (3) Carcharodontosauridae mesial as Carcharodontosauridae lateral or as Tyrannosauroidea lateral, (4) Non-abelisauroid Ceratosauria lateral as Metriacanthosauridae lateral or as Tyrannosauroidea lateral, (5) Abelisauridae mesial as Abelisauridae lateral or as Allosauridae lateral, (6) Metriacanthosauridae lateral as non-abelisauroid Ceratosauria lateral, (7) Dromaeosauridae mesial as Dromaeosauridae lateral, (10) Non-averostran Neotheropoda lateral as Dromaeosauridae lateral, (11) Non-abelisauroid Ceratosauria mesial as Allosauridae lateral, (12) Neovenatoridae lateral as Non-spinosaurid Megalosauroidea lateral or as Dromaeosauridae lateral, (13) Basalmost
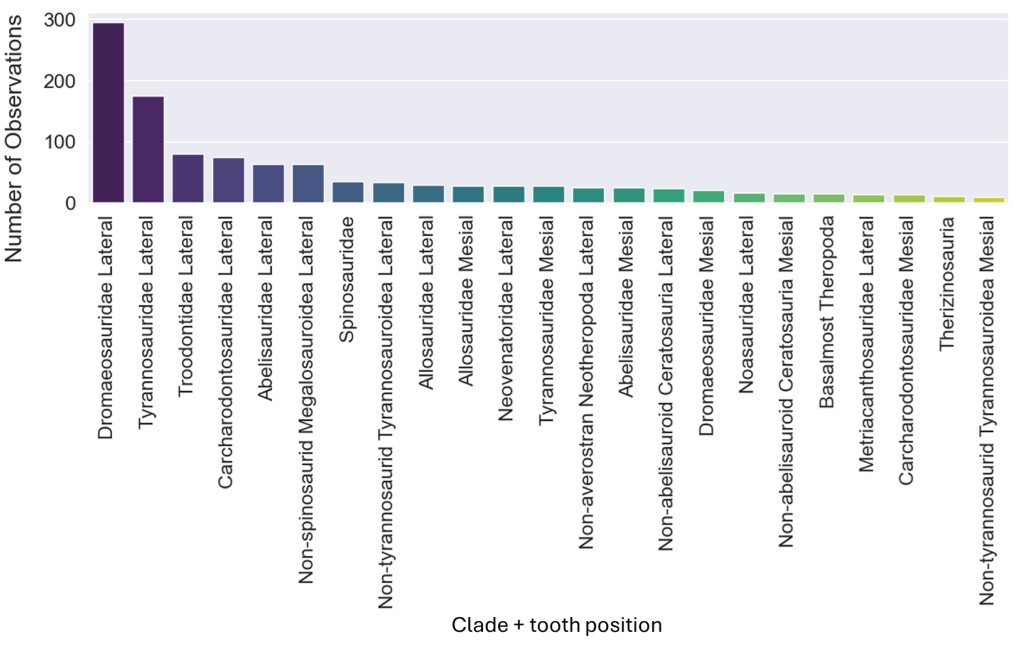

**Figure 9** Number of observations for each of the twenty-three combinations of clades and tooth positions.

Theropoda as Dromaeosauridae lateral, (14) Abelisauridae mesial as Abelisauridae lateral or as Allosauridae mesial (Fig. 11).

Even-though the confusion matrix only shows the predicted class, the three most probable classes were evaluated in the cases were the prediction was incorrect for the whole dataset, resulting in a total of 97.9% correct predictions.

Similar to the genus-level models, the most important variable for these predictions was CBR, with a mean importance of 0.14 (Fig. 12), followed by DC with a mean importance of 0.12 (Fig. 12).

The first branches of the decision trees show that the first criteria to do the classification was if log-scaled CH is lower or equal to −0.218, the second decision was if (1) log-scaled CBR was lower or equal to 1.477, (2) log-scaled DC lower or equal to 0.789 (Fig. 13).

The best performing model was used to classify the set of isolated theropod teeth from Guimarota fossil site and all of the teeth were attributed to their corresponding classification in the three most probable classifications, except Tyrannosauridae mesial (Table 6)

## DISCUSSION

### Classification algorithms

In this study, RF and GB were the most robust models, presenting the best averaged performance and consistency across genus and higher taxonomic-level classifications regardless of oversampling- log-scaling or standardization. Their tree-based structure, that partitions data through feature thresholds rather than distance metrics, allows a more robust and consistent performance of these models, even when data scale changes.

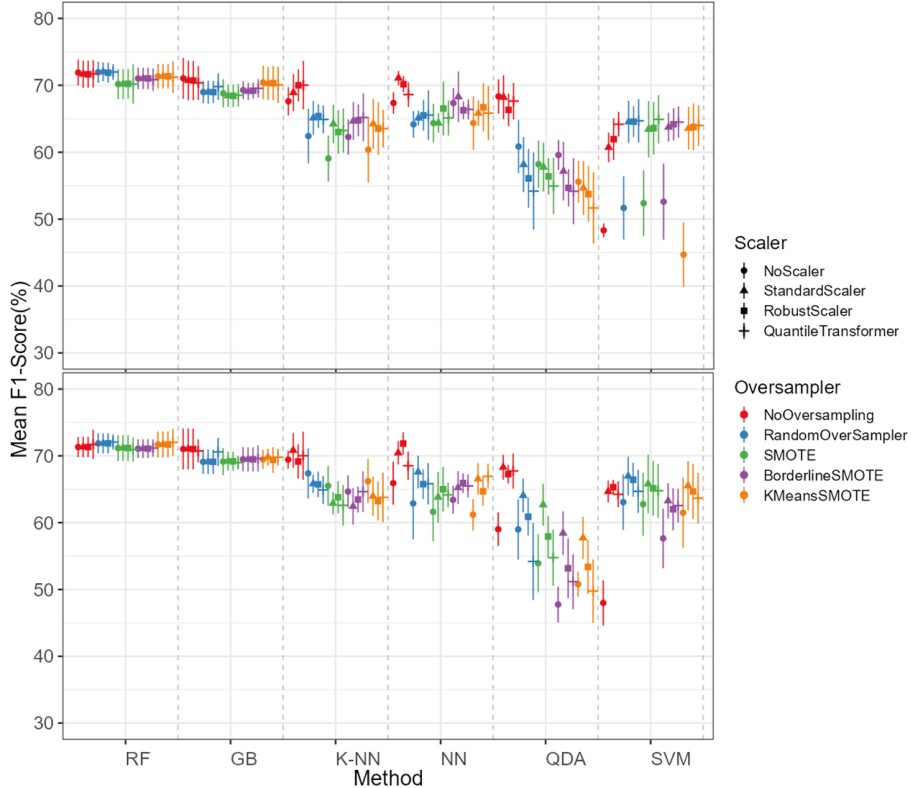

**Figure 10** **F1-score values for each combination model, oversampling and scaler.** The top plot corresponds to the results of the models using the non-log-scaled variables and the bottom plot corresponds to the model results using the log-scaled variables. The dotted line corresponds to the separation of the different methods.

StandardScaler presented better averaged metrics for most of the classification tasks, either when using or not log-scaled variables and for the classification at both genus and higher taxonomic levels. The results obtained show that this method can effectively handle outliers and non-normal data distributions, since it standardizes features by removing the mean and scaling to unit variance.

Models like SVM and NN presented more variability, as their performance depends on standardized inputs to avoid skewed distance metrics, benefiting significantly from combined standardization and oversampling. An example of the influence of standardization and oversampling on the model results is the SVM for the genus classifier using the non-log-scaled variables, where the F1- increases 36.7%, from 37.6% (not using any oversampling or standardization technique) to 74.3% (using the BorderlineSMOTE method and the RobustScaler).

Despite the goal of addressing class imbalance, oversampling provided limited improvements, suggesting potential overfitting, where synthetic samples may overly tailor models to the training set without generalizing well to the test data. Tree-based models easily adapted to different scenarios, presenting good performance results without the need of oversampling due to their natural handling of class imbalance through ensemble
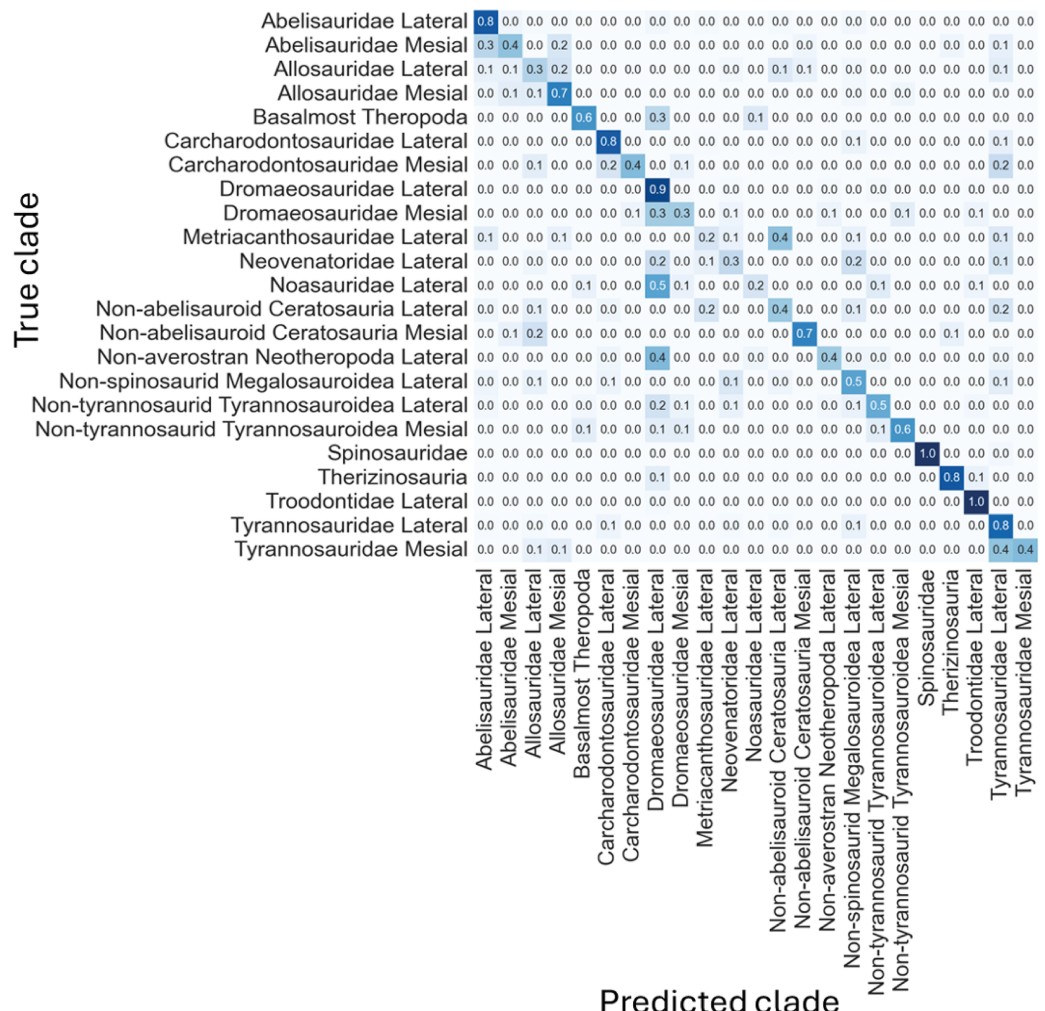

**Figure 11** Normalized confusion matrix showing the mean proportion of predicted labels for each class.

structures. One of the possible reasons why oversampling did not increase most of the models' performance is that the oversampled class will be less variable than a class that consists entirely of real data, increasing the possibility of the model not being able to generalize to new data. Standardization played a key role in improving distance-sensitive models like SVM, which relies on balanced feature scales to establish unbiased class boundaries effectively. These findings highlight that while oversampling can benefit SVM and similar models, tree-based models, such as RF and GB, perform optimally across variable scales without oversampling, suggesting that these latter models would be the preferred classifiers in imbalanced datasets.

While using the permutation importance, the two variables with more weight for the predictions in the case of the datasets used in this work were: DC and CBW for the genus level classification; CBR and DC for the higher taxonomic level classification. These findings suggest that variations in denticle density can help differentiate between

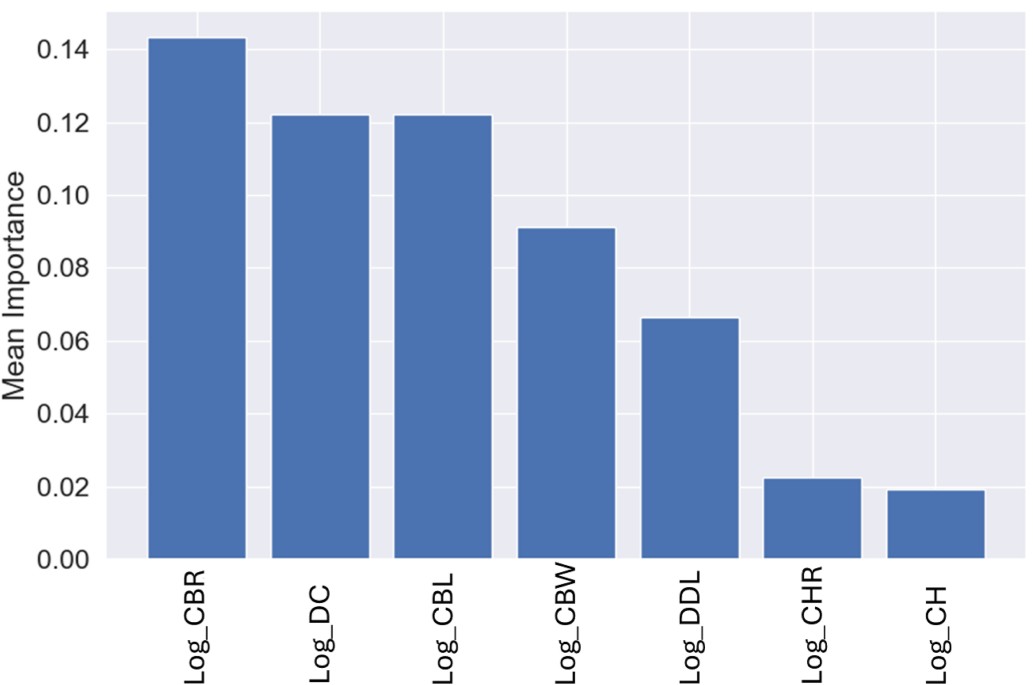

**Figure 12** Mean permutation importance of the different variables for the best classification model.

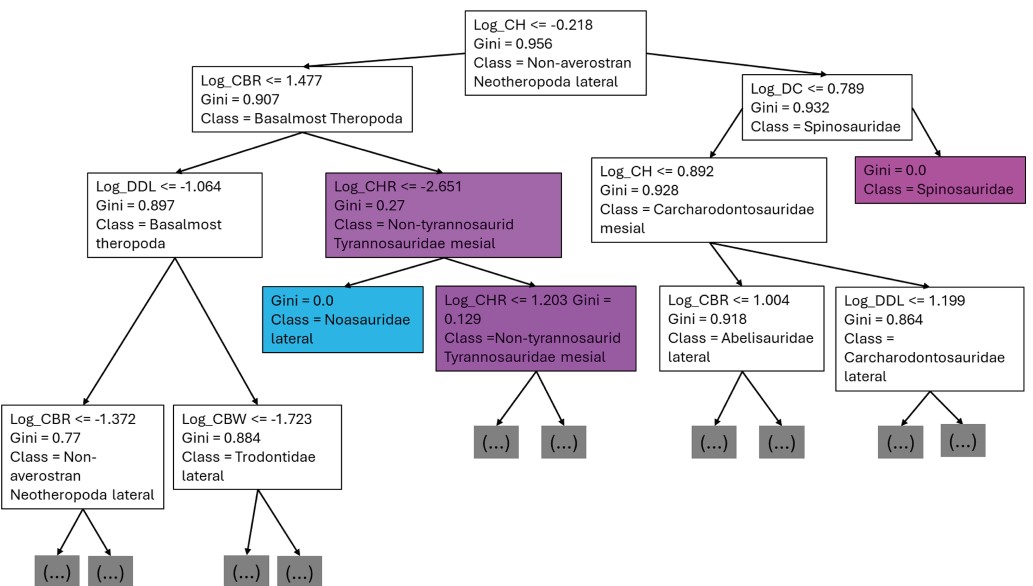

**Figure 13** First three branches of decisions of the first decision tree of the best model.

**Table 6   Initial classification and the three most probable classifications using the best performing model.**

| Inventory number | Most probable label | Second probable label | Third probable label |
|---|---|---|---|
| MG 27805_D193 | Dromaeosauridae lateral | Troodontidae lateral | Noasauridae lateral |
| MG 27782 | Troodontidae lateral | Dromaeosauridae lateral | Dromaeosauridae mesial |
| MG 27768 | Dromaeosauridae lateral | Non-averostran Neotheropoda lateral | Non-tyrannosaurid Tyrannosauroidea lateral |
| MG 27808_D118 | Dromaeosauridae lateral | Dromaeosauridae mesial | Non-averostran Neotheropoda lateral |

several genus and clades, making it a valuable trait for classification of isolated theropod teeth as was previously highlighted (*Hendrickx et al., 2019*). Since CBR is a measure of crown shape proportions, it could capture broader evolutionary trends and ecological adaptations, which are shared among related higher taxonomic level groups. Therefore, theropod higher taxonomic levels with specific dietary niches, such as the robust teeth of apex predators *versus* the narrow, conical teeth of piscivorous taxa, might show distinct patterns in CBR that differentiate groups like Tyrannosauroidea and Spinosauridae. On the other hand, CBW might have provided finer distinctions within higher taxonomic levels, considering subtle variations in tooth base robustness related to feeding mechanics or dietary specialization unique to specific genera.

In the results section it was possible to see quite common misclassifications between some theropod clades such as Noasauridae lateral and Dromaeosauridae lateral or Allosauridae mesial and Allosauridae lateral (Fig. 14). This misclassification often happens due to similarities in the overall teeth morphology of these theropod clades. *Hendrickx, Mateus & Araújo (2014)* discusses that Megalosauridae, Ceratosauridae, Abelisauridae, Allosauridae, and Neovenatoridae show considerable overlap, and that significant overlapping also exists for Tyrannosauridae and Carcharodontosauridae, making the distinction between teeth of megalosaurids from the teeth of other similarly sized theropods particularly difficult. The authors also discussed the presence of overlap among different allosauroid taxa, including *Allosaurus* and *Acrocanthosaurus*, showing that the distinction between mesial and lateral teeth is not that clear in this clade. *Hendrickx, Mateus & Araújo (2014)* also divided theropod teeth into four morphospace areas: (1) taxa with small teeth and large denticles (Troodontidae); (2) taxa bearing relatively small teeth and small denticles (non-neotheropod Theropoda, Coelophysoidea, Noasauridae, and Dromaeosauridae); (3) taxa possessing large teeth and minute denticles (Spinosauridae); and (4) ziphodont taxa having relatively large teeth and large denticles (non-noasaurid Ceratosauria, Megalosauridae, Allosauroidea, and Tyrannosauridae). The study argues that overlapping exists between each of these areas, and clades bearing small teeth/denticles and large teeth/ denticles show considerable overlap.

It is important to note that only a small subset of the available variables were used in this analysis, since we only used the variables with less than 15% missing values. While this ensured that the data used was consistent and widely applicable across most teeth, it introduced a possible limitation. Variables that can be critical for identifying certain genus

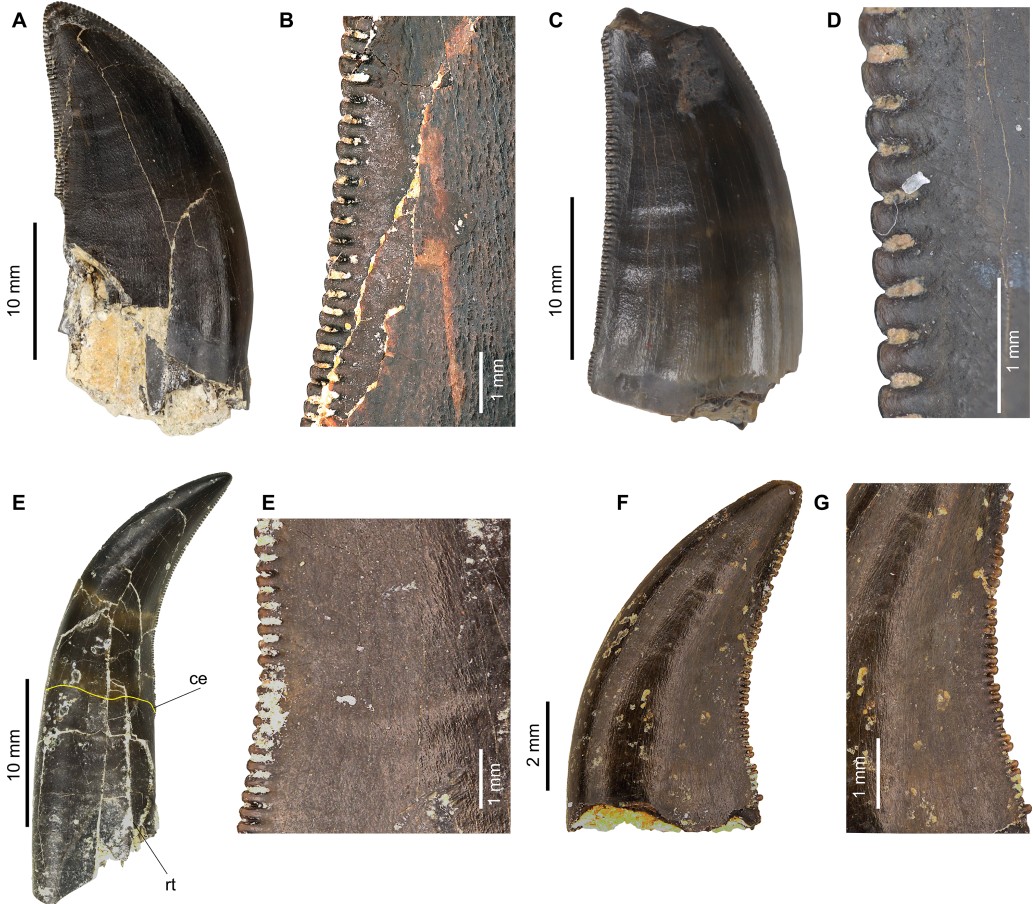

**Figure 14 Comparison between some of the most misclassified isolated theropod teeth.** (A) MNHN/UL.AND.24 interpreted as lateral tooth of *Allosaurus*, by E Malafaia, F Batista, B Maggia, CS Marques, F Escaso, P Dantas, F Ortega (2025, unpublished data) in labial view (B) detail of the denticles from the central section of the distal carina of MNHN/UL.AND.24; (C) MNHN/UL.EPt.019 interpreted s a mesial tooth of *Allosaurus* by *Malafaia et al. (2024)* in labial view) (D) detail of the denticles from the central section of the distal carina of MNHN/UL.EPt.019; (E) MNHN/UL.AND.105 interpreted as a Non-tyrannosaurid Tyrannosauroidea, by E Malafaia, F Batista, B Maggia, CS Marques, F Escaso, P Dantas, F Ortega (2025, unpublished data) in labial view. (F) Detail of the denticles from the central section of the distal carina of MNHN/UL.AND.105; (G) MNHN/UL.AND.104 interpreted as belonging to a Dromaeosauridae, by E Malafaia, F Batista, B Maggia, CS Marques, F Escaso, P Dantas, F Ortega (2025, unpublished data) in lingual view; (H) detail of the denticles from the central section of the distal carina of MNHN/UL.AND.104. Abbreviations: ce, cervix and rt, root.

but are incomplete or difficult to measure in other genera are excluded. This restriction could lead to an increased difficulty of the methods to correctly identify genera that are dependent on other variables. This increases the need for researchers to evaluate the reasonability of the predictions, and the importance of evaluating the three most probable classes instead of looking just for the most probable prediction.

## Application of ML classifiers to new findings of isolated theropod teeth

The results obtained from our study can be directly applied to the analysis of new findings of isolated theropod teeth, using the trained models and standardizations, or using the same methodologies but training the models from scratch. On the other hand, it is important to note that the training contains a limited number of observations per genus, making it difficult for the models to not overfit to the dataset. Therefore, care should be taken if these models are applied to unseen data from different populations. The following guidelines aim to assist researchers in effectively using the techniques tested in the study of other fossil records.

When classifying isolated theropod teeth, an initial step involves selecting suitable classification algorithms. Tree-based models, such as RF and GB, are robust options, especially when working with raw datasets where standardization and oversampling are not applied. These algorithms rely on decision trees that split data based on feature thresholds, making them well-suited for non-standardized data since they are not sensitive to feature scaling. When applying pre-trained models, it is essential to replicate the pre-processing steps used during training to maintain consistency. If standardization was applied during training, all features must be standardized in the same way, using identical parameters, to ensure the model's effectiveness.

When classifying paleontological data, oversampling techniques should be considered if the dataset exhibits class imbalance, a common issue due to over-representation of certain taxa. In our study, oversampling notably improved the performance of SVM models, with SMOTE and RandomOversampler proving most reliable. Specifically, the best model for both genus and higher taxonomic level classification used RandomOversampler. This technique is valuable in balancing datasets where teeth from more abundant taxa, such as *Tyrannosaurus* or *Saurornitholestes* in the case of the dataset used in this study, overshadow those of less common genera. Implementing RandomOversampler can thus enhance classification performance analyzes. However, oversampling should be avoided when using pre-trained models exclusively for inference on new data, as altering the class distribution could degrade model accuracy.

The decision to apply log-scaling should be guided by the dataset's characteristics and the model's sensitivity to skewed distributions. RF and GB models typically perform similarly on log-scaled and non-log-scaled data, as they are less sensitive to distribution shapes. However, log-scaling can be advantageous for algorithms such as SVM and NN, which may benefit from more symmetrically distributed features. Log-scaling, especially when combined with standardization, can stabilize variance and improve model performance when features vary across several orders of magnitude or exhibit skewed distributions.

## Application of the methodology in the sample of isolated theropod teeth from Guimarota fossil site

The four isolated theropod teeth mentioned in the methodology section were used to test the classifiers predictions for unseen isolated theropod teeth. The results of the classification using the best models showed that the first three most probable predictions were similar to the original classification. MG 27808_D118 was attributed to *Richardoestesia* by *Zinke*

*(1998)* and the results obtained on our classifier models recovered it as *Saurornitholestes* and *Richardoestesia* as the firsts and second highest probabilities (Table 5). The results for the other specimens are somewhat more uncertain, with MG 27805_D193 being classified as *Troodon* and *Saurornitholestes*, MG 27782 as *Troodon* and *Dilong*, and MG 27768 as *Masiakasaurus* and *Saurornitholestes* as the first and second highest probabilities in the model based on the genus-level dataset. Based on the higher-taxonomic level dataset, MG 27805_D193 is classified as Dromaeosauridae and Troodontidae, MG 27782 as Troodontidae and Dromaeosauridae, and MG 27768 as Dromaeosauridae and non-averostran Neotheropoda (Table 6). MG 27805_D193 is associated to a labelled with the identification *Ceratosaurus*-like. An isolated tooth from Guimarota was described by *Rauhut, Martin & Krebs (2000)* as belonging to a *Ceratosaurus*-like theropod. However, this specimen has the distal margin almost straight with the apex positioned at the level or more mesial to the most distal end of the distal carina whereas the crown of MG 27805_D193 is strongly recurved, with the mesial margin strongly convex and the distal one concave. Besides, it has an eight-shaped basal cross-section due to the presence of small depressions centrally positioned in the labial and lingual surfaces, which is a character typical for the lateral dentition of dromaeosaurids. Also the presence of much larger distal than mesial denticles (DSD$I$ > 1.2) is also a feature common in most dromaeosaurids. Therefore, the identification obtained based on the classifier models is compatible with the combination of features shown by the specimen suggesting that it probably belongs to a dromaeosaurid theropod.

The specimens MG 27782 and MG 27768 are associated with labels identifying them to tyrannosaurid theropods. Some isolated teeth attributed to tyrannosauroids were described in the Guimarota fossil site by *Zinke (1998)* and *Rauhut, Martin & Krebs (2000)*. However, these previously described specimens correspond to teeth from a more mesial position in the tooth row compared to the two here analyzed. Both MG 27782 and MG 27768 show the distal carina strongly displaced to the lingual surface, which is a feature shared with several tyrannosauroid taxa but also with several other theropods, including dromaeosaurids (*e.g.*, *Hendrickx, Mateus & Araújo (2015)*). Therefore, the taxonomic interpretation of these specimens needs to be reviewed based on more uptodate methodologies. An attribution of these isolated teeth to both non-tyrannosaurid tyrannosauroid and dromaeosaurid theropods as suggested by the results of the analyses here performed cannot be excluded at this moment.

## CONCLUSIONS

The combination of data standardization, oversampling, and classification algorithm choice was critical for building reliable machine learning models. Standardization balanced variable contributions, while oversampling addressed class imbalance. Among classifiers, RF provided the best fit for classifying isolated theropod teeth, showing robustness to standardization, oversampling and log-scaling, with KMeansSMOTE being the oversampling method used by the best performing model to classify the teeth at genus level. Future research should explore incorporating non-morphometric features (*e.g.*, tooth

ridges) and alternative variable selection methods, as important variables may have been excluded during pre-processing. This study offers a framework for identifying isolated theropod teeth and their position based on morphometric variables. While trained models and standardizations are available for future studies in Github, they should be applied with caution, mindful of limitations such as taxa-specific training and dataset biases.

## ACKNOWLEDGEMENTS

We are grateful to C. Hendrickx for making the original dataset publicly available. CSM would like to thank Afonso Barrocal for the early comments on the first draft of this research. We also acknowledge the Phylopic website for the theropod silhouettes, and thank Jagged Fang Designs, Pranav Iyer and Tasman Dixon for sharing their original artworks on this website. We also thank Judite Alves and Roberto Keller (Museu Nacional de História Natural e da Ciência, Universidade de Lisboa) for the access to the specimens and to the equipment and software for taking the pictures and measurements of the specimens. We also thank Jorge Sequeira and Ruben Dias (Museu Geológico of Laboratório Nacional de Engenharia e Geologia (LNEG- Lisboa)) for access to the specimens from Guimarota fossil site. We also thank Bruno C. Silva (Sociedade de História Natural, Torres Vedras) for the access to the specimens housed at the collection from Sociedade de História Natural.

### Funding

This work was supported by Portuguese government funds through FCT - Fundação para a Ciência e Tecnologia under the doctoral scholarship FCT/CEAUL (UI/BD/154258/2022), the individual contract CEECIND/01770/2018 (https://doi.org/10.54499/CEECIND/01770/2018/CP1534/CT0004), CEAUL's strategic projects: UID/00006/2025 and UIDB/00006/2020 (https://doi.org/10.54499/UIDB/00006/2020) for the APC, and FCT, I.P./MCTES through national funds (PIDDAC): UID/50019/2025, UIDB/50019/2020 (https://doi.org/10.54499/UIDB/50019/2020) and LA/P/0068/2020 (https://doi.org/10.54499/LA/P/0068/2020). Emmanuel Dufourq was funded by a grant from the Carnegie Corporation of New York (provided through the AIMS Research and Innovation Centre). There was no additional external funding received for this study. The funders had no role in study design, data collection and analysis, decision to publish, or preparation of the manuscript.

### Grant Disclosures

The following grant information was disclosed by the authors:
FCT - Fundação para a Ciência e Tecnologia under the doctoral scholarship FCT/CEAUL (UI/BD/154258/2022).
The individual contract CEECIND/01770/2018.
CEAUL's strategic projects: UID/00006/2025 and UIDB/00006/2020.
APC, and FCT, I.P./MCTES through national funds (PIDDAC): UID/50019/2025, UIDB/50019/2020 and LA/P/0068/2020.
Carnegie Corporation of New York.

## Competing Interests

The authors declare there are no competing interests.

## Author Contributions

- Carolina S. Marques conceived and designed the experiments, performed the experiments, analyzed the data, prepared figures and/or tables, authored or reviewed drafts of the article, and approved the final draft.
- Emmanuel Dufourq performed the experiments, analyzed the data, authored or reviewed drafts of the article, and approved the final draft.
- Soraia Pereira analyzed the data, authored or reviewed drafts of the article, and approved the final draft.
- Vanda F. Santos analyzed the data, authored or reviewed drafts of the article, and approved the final draft.
- Elisabete Malafaia conceived and designed the experiments, performed the experiments, analyzed the data, authored or reviewed drafts of the article, and approved the final draft.

## Data Availability

The code are available at Github and Zenodo:

- https://github.com/MCarolinaMarques/Enhancing-Fossil-Classification-A-Guide-to-Machine-Learning-Methods-for-Theropod-Teeth - contains all data, code, results and models.

- Carolina S. Marques. (2025). MCarolinaMarques/Enhancing-Fossil-Classification-A-Guide-to-Machine-Learning-Methods-for-Theropod-Teeth: Code and data for the article "Enhancing Fossil Classification: A Guide to Machine Learning Methods for Theropod Teeth" (Code). Zenodo. https://doi.org/10.5281/zenodo.14907721.

The metric results for all the models are available at ShinyApp: https://ml4paleontology.shinyapps.io/teethshinyapp.

The published dataset containing the measurements are available at Zenodo: S. Marques, C., Dufourq, E., Pereira, S., Faria dos Santos, V., & Malafaia, E. (2025). Isolated theropod teeth from Guimarota fossil site presented in "Enhancing the classification of isolated theropod teeth using machine learning: a comparative study" [Data set]. In Enhancing the classification of isolated theropod teeth using machine learning: a comparative study. Zenodo. https://doi.org/10.5281/zenodo.14632904.

## Supplemental Information

Supplemental information for this article can be found online at http://dx.doi.org/10.7717/peerj.19116#supplemental-information.

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
