# Peer review of "Enhancing the classification of isolated theropod teeth using machine learning: a comparative study"

_PeerJ, doi:10.7717/peerj.19116_

## Round 0.1 · original submission · Major Revisions

This is a timely manuscript, building upon published datasets and methods. The reviewers all note some areas that should be addressed in revision, particularly around the methodology and sample used here.

- Reviewer 2 suggests additional illustrations; I agree that a basic diagram of a theropod tooth (or teeth) is highly desirable. The reviewers as a whole provide several other suggestions for modified or additional figures, which should be considered in your review.
- All three reviewers highlighted several concerns with the Epoch variable as part of the analysis; this definitely should be addressed, likely through removing the variable. Similarly, use of a ratio alongside its raw variables introduces the multicollinearity problem, as noted by Reviewer 3.
- A number of other suggestions are provided, all of which should be incorporated or rebutted as appropriate.

Given the scope and breadth of the suggested revisions, I have suggested "major revisions" (with likely re-review) before resubmission.

·

Basic reporting

Dear authors,

it is worthy to point out and test the importance of normalisation and oversampling for machine learning approaches. I have a few comments, of course:

In general, I advice to be more critical and prudent in the discussion when it comes to the reliability of the approach. One important thing to remember is that palaeontological data is never independent. The training set will come from a very limited number of populations per species (one or very few). Consequently, overfitting cannot be really avoided: If the method is applied to unseen data from a different population, it will perform worse.

I also like to note that oversampling cannot fully get rid of the problem of class imbalance. The oversampled class will necessarily be less variable than a class that consists entirely of real data, which should create a bias towards the latter, no?

In particular, I am a bit concerned about the use of the variable "Epoch" (i.e., geological time) to train the models, for two reasons:

1) Shortcut learning is more likely to happen. Let's say that, for a particular epoch (let's say, Late Triassic), there are only two dinosaur taxa in the dataset, and both differ in size. A model trained on this data might not take tooth shape into account at all in this case, but just relies on epoch and overall size as shortcuts. If used on new data of different taxa, the method may fail horribly.

2) Circularity. Let's contemplate about how palaeontologists will use such methods. They will classify teeth from new/different localities; these records go into databases, which are then used to model the distribution of dinosaur taxa in space and time. Now, let's assume there are many records in the database identified using your method. The data could show a sharp faunal change at the boundary between Triassic and Jurassic, for example, which palaeontologist then interpret as evidence for a mass extinction. But in fact this is just an artefact, because the data was biased due to the machine learning method using geological time as a criterion to start with!


The "Discussion" is a bit brief and seems to omit an important point: Only a small subset of the available variables were used (those with less than 16% missing values), if I understand the "Methods" section correctly. This means that the analysis is restricted to variables that can be measured in most teeth. However, it can be that there are variables that capture crucial features in some species, but cannot be applied to other species. As a result, there would be species that cannot be identified by the method, because they would require variables that are removed just because that cannot be measured in the other species.


Minor comments:

Check capitalisation in references. Proper names have to be capitalised (e.g., Dinosauria, Theropoda, Morocco)

l. 112: The plural of "genus" is "genera".


Jens N. Lallensack

Experimental design

see above

Validity of the findings

see above

Additional comments

see above

·

Basic reporting

The authors here present a machine learning study on the classification of theropod teeth primarily based on their morphometric measurements. A large, in vertebrate paleontological scale, dataset has been compiled and various machine learning methods have been tested. However, I believe this study can be improved in at least the following four aspects:

1. How machine learning and geometric morphometric studies have been conducted on theropod teeth, especially how the dataset scale and coverage change/increase through time since one focus of this study is dataset imbalance.
2. Experimental design, please see the next section.
3. Illustrations. Multiple figures are shown along with the manuscript to show the results, however, current visualization is not enough to help readers understand the purpose and scope of this study. There are several figures that should be added:
a. An example isolated theropod tooth with all the morphometric variables. It can be modified from previous studies.
b. Some example teeth from different species to show the morphometric/morphological disparity. The teeth should “look differently” so that they can be somehow classified.
c. Visualization of classification results. For example, if SVM model is used with rbf kernel (from the code), a 2D or 3D plot with the most important variables in which different groups of theropod teeth occupying different area. It would be helpful to see how different ML method divide the abstract morphometric space and possibly the same method with different parameters (e.g. rbf, linear, sigmoid, poly……kernels in SVM).
d. Comparison between easily confused teeth. It would be ideal to directly see why two categories are confused, for example Noasauridae lateral vs Dromaeosauridae lateral, Tyrannosaurpidea mesial vs Tyrannosaurpidea lateral, etc.
4. Test on new specimens. To myself, the most important part of this study is to identify new isolated theropod teeth and position, but I have not seen any of this from the manuscript. Because all data mentioned here have been used in the training process, testing the trained model on totally new specimen data would greatly increase the validity of this study.

Experimental design

1. Although six selected methods are briefly introduced in Data modeling section, their detail is lacking. For example, what are the number of estimator, max features, max depth, etc of 1) random forest? What is the architecture of 5) neural network, or even at least how many layers are there? What kernel is used in 4) SVM? All six model should be more clearly explained.
2. It is ideal but not quite possible to train model on all theropod species, but it is necessary to see what is the coverage of the sampled dataset, not only taxa coverage but also other factors that may influence classification results like body size, eco-type, biogeography, etc.
3. Following point 2, what is the phylogenetic framework used here, there are many ambiguous groups like ‘early-branching Theropoda’ (while lacking the position information) or ‘Non-averostran Neotheropoda lateral’. My suggestion is to add a phylogeny tree showing both your definition of each group/clade and the taxonomic coverage.
4. The variable 21) geological age is named as Epoch, but from the code I find it is quite confusing for two reasons: a) there is no official Middle Cretaceous; b) the coding shows the younger the age is, the larger the number is, but why? And even a log-transfor is performed on the number-coded geological age. More importantly, figure 4 indicates that the Epoch is the most important variable, what if we flip the coding order or simply use absolute age instead of 0-6. The geological age variable can be abandoned, but the performance is likely getting worse; or coded in a more reasonable way; or still coded like this but provide enough explanation to show it is more superb than other coding method.
5. Imbalance sampling. I understand the authors have tried various methods to avoid its influence and the difficulties of collecting fossil data. A simple but definitely doable way to test this is to do experiments on all samples but the two most abundant categories, Saurornotholestes lateral and Tyrannosaurus lateral.
6. Why do Eoraptor, Sinraptor, and Monolophosaurus do not have lateral and mesial?

Validity of the findings

Current results are more like preliminary studies, I suggest the authors try to do extra experiments as suggested in 2. Experimental design, especially 2. testing brand new specimens. If the model performance still holds, we can confirm the validity of this study.

Additional comments

Line 57, posterior → later
Line 64-47, imbalance itself means over-representation and underrepresentation
Line 104, no need to repeat full name
Line 107, refers for modified SMOTE methods?
Line 147, Palaeontology → palaeontology
Line 179, since ‘geological age’ is not really morphometric, leave just variables here

·

Basic reporting

Thank you for allowing me the opportunity to review this paper. The use of machine learning to help classify isolated dinosaur teeth is at an early stage, and anything that can help gain more acceptance for this powerful method is welcome.

The authors aim is to compare different machine learning models, using morphometric data from published datasets, to assess how accurately they can classify isolated theropod teeth. In addition, the authors quite correctly highlight that one issue with such datasets is around class imbalance and they look at different methods to address this.

The paper is well written and the background study comprehensive. I do however have a number of minor issues with the background review, which I feel should be corrected and relate to the previous studies I have contributed to.

Lines 107 to 109 state that the Wills et al. (2020) study only used log-scaling to the variables rather than feature scaling. This is incorrect and in fact feature-scaling was used and is clearly mentioned in the methods section for that paper. It would also be useful to mention that our results suggested that the oversampling technique we used (SMOTE) was actually sub-optimal and negatively affected the classification.

Lines 117 to 119: state that the Wills et al. (2023) study undertook no standardisation of data. This is incorrect. The study used the same methodology as the Wills et al. (2020) study and is clearly documented. The authors also point out that we did not undertake any oversampling, which is correct, but I would note that the rationale for not doing so is made clear in that paper.

Lines 125 to 127: same comment as per lines 117 to 119 for the Hendrickx et al. (2023) study.

I have one other comment on the background. In lines 169 to 172 the authors state that a key contribution of this study is that the models, and not just the data have been made available. It is great to see this, and it is also positive that the code is available in Python. However, I fail to see the difference between this and the supplemental information provided in Wills et al. (2020, 2023) and Hendricks et al. (2023) where again both data and code (R) were provided.

My only other comments on the basic reporting are:

I think the figures need looking at to present the results in a clearer way. It would be more useful to have a table summarising the results in conjunction with figures 2 and 6. I think figures 4 and 8 could be combined, the same for figures 1 and 5.

Much of the methods section is taken up describing the different scaling techniques, modelling and metrics used. I think that the detailed equations and descriptions of the equations given in the text detract from the manuscript. These were not derived for this study and are effectively standard equations from other works. This section should be comprehensively edited and, if required, the equations placed in a supplementary file.

Experimental design

Unfortunately the study, as it stands, falls down on the experimental design although this could be corrected quite easily.

The issue is around the selection of variables used in the model. The authors state they use the following:

Epoch, DC, DDL, CBW, CBL, CBR and CH (lines 191 & 192) although I note that figures 4 and 8 suggest an additional variable, CHR, was also used.

My major issue is including the chronostratigraphic variable Epoch in the analysis. This variable is simply showing that the model can differentiate on the currently known chronostratigraphic provenance of that taxa. I would go further and suggest that, as their results currently show that Epoch is the most important variable, a model containing this would hinder efforts to identify isolated teeth from a clade that falls outside its currently known stratigraphic boundaries. I strongly advise the authors to re-run the analysis using morphometric measurements only unless they can strongly justify this in their methods and results.

My other issue is around the use of the ratio variables CBR and CHR in the analysis. CBR s a ratio of CBW / CBL and CHR a ratio of CH / CBL. In the model the authors include both the original raw variables and the ratio variables. This could lead to issues with multicollinearity as the ratios are often highly correlated with the original variables – although I accept that different models are less sensitive to this issue.

I would like to see the authors clearly state their rationale and justification with using both ratio and raw variables in their analysis.

Validity of the findings

Unfortunately as I have major issues with the experimental design around the choice of variables used in the analysis I cannot comment on the validity of their finding s until either the analysis is re-run or a strong justification is provided.

Additional comments

I think this study is still of value, notwithstanding my issues around the current experimental design, which I why I am suggesting a major revision. The evaluation of different methods to improve training data is important and I hope that the authors can address my concerns.

---

## Round 0.2 · Minor Revisions

Thank you for your thorough attention to the comments from the reviewers; the manuscript is much improved, as they noted following a second round of review. At this point, there are just a handful of minor points raised by the reviewers that you should address, please. I also found a few very minor issues to resolve (mostly typos).

MINOR COMMENTS FROM EDITOR
- line 104: change "genus" to "genera" (because it is plural)
- line 108: Split the sentence; "...were employed. The authors only log-scaled the variables..." (and remove the extra "only")
- line 144: list as (Hendrick et al., 2023; Wills et al., 2020).
- Figure 1: CC-BY-SA is not compatible with the PeerJ license, in my understanding (journal staff may be able to provide clarification). Please confirm if you may use these images, or if you must find alternates for Figure 1C and 1E.
- line 274: you may remove "Friedman" from the parentheses, because it is already in the sentence "...score by Friedman (2011),..."

·

Basic reporting

All my concerns with the first submission have been addressed. The variable "epoch" was removed from the analysis as requested, additional tables were added, and additional information on the limitations of the method was included as requested. Publishable as is now.

Experimental design

see above

Validity of the findings

see above

Additional comments

see above

·

Basic reporting

After a major revision, the authors have presented a much improved manuscript along with better designed experiments and plenty of new illustrations. The authors have revised this study following what the two other reviewers, the editor, and I suggested to show the application of ML methods in paleontological study. The revised version provides more background for fresh readers. I am delighted to recommend the publication of this study!

Experimental design

No comment.

Validity of the findings

No comment.

Additional comments

The caption of figure 14 has two misspelling "Allosauros", should be "Allosaurus".

·

Basic reporting

Thank you for allowing me to review this resubmitted manuscript. I thank the authors for addressing all my concerns raised in my initial review.

I only have one comment regarding the revised manuscript. In your response to the reviewers you state that:
"We believe that the difference is that we can use the trained model that we obtained in our work instead of only following the code. The model is made available in a .pth file which can be loaded in Python and there is no need to train a new model, just apply the model that was trained in this analysis to the new data, following the standardization also obtained by our methodology. Everything will be made available upon acceptance for publication in the Github page, including an easy-to-follow Python file were the applications of the trained model is used to classify the new isolated theropod teeth that were added to the manuscript (four isolated theropod teeth from the Kimmeridgian (Upper Jurassic) Guimarota fossil site)."

I cannot see any reference to the Github repository or to the .pth file in the manuscript and suggest making the availability of this a little clearer.

Apart from that I have no more concerns regarding the manuscript.

Experimental design

no comment

Validity of the findings

no comment

Additional comments

no comment

---

## Round 0.3 · accepted · Accept

Thank you for your close attention to the final round of changes; the manuscript is ready to move forward!